# Machine learning for spatial stratification of progressive cardiovascular dysfunction in a murine model of type 2 diabetes mellitus

**Andrya J. Durr**[1,2¤], **Anna S. Korol**[3], **Quincy A. Hathaway**[1,2,4], **Amina Kunovac**[1,2,4], **Andrew D. Taylor**[1,2], **Saira Rizwan**[1,2], **Mark V. Pinti**[2,5,6], **John M. Hollander**[1,2]*

1 Division of Exercise Physiology, West Virginia University School of Medicine, Morgantown, West Virginia, United States of America, 2 Mitochondria, Metabolism & Bioenergetics Working Group, West Virginia University School of Medicine, Morgantown, West Virginia, United States of America, 3 Department of Neuroscience, Rockefeller Neuroscience Institute, West Virginia University School of Medicine, Morgantown, West Virginia, United States of America, 4 Center for Inhalation Toxicology (iTOX), West Virginia University School of Medicine, Morgantown, West Virginia, United States of America, 5 West Virginia University School of Pharmacy, Morgantown, West Virginia, United States of America, 6 Department of Physiology and Pharmacology, West Virginia University School of Pharmacy, Morgantown, West Virginia, United States of America

¤ Current address: Department of Social and Behavioral Sciences, West Virginia University School of Public Health, Morgantown, West Virginia, United States of America

* jhollander@hsc.wvu.edu

**Data Availability Statement:** All datasets used for machine learning analyses and related code have been made publicly available and can be accessed

## Abstract

Speckle tracking echocardiography (STE) has been utilized to evaluate independent spatial alterations in the diabetic heart, but the progressive manifestation of regional and segmental cardiac dysfunction in the type 2 diabetic (T2DM) heart remains understudied. Therefore, the objective of this study was to elucidate if machine learning could be utilized to reliably describe patterns of the progressive regional and segmental dysfunction that are associated with the development of cardiac contractile dysfunction in the T2DM heart. Non-invasive conventional echocardiography and STE datasets were utilized to segregate mice into two pre-determined groups, wild-type and *Db/Db*, at 5, 12, 20, and 25 weeks. A support vector machine model, which classifies data using a single line, or hyperplane, that best separates each class, and a ReliefF algorithm, which ranks features by how well each feature lends to the classification of data, were used to identify and rank cardiac regions, segments, and features by their ability to identify cardiac dysfunction. STE features more accurately segregated animals as diabetic or non-diabetic when compared with conventional echocardiography, and the ReliefF algorithm efficiently ranked STE features by their ability to identify cardiac dysfunction. The Septal region, and the AntSeptum segment, best identified cardiac dysfunction at 5, 20, and 25 weeks, with the AntSeptum also containing the greatest number of features which differed between diabetic and non-diabetic mice. Cardiac dysfunction manifests in a spatial and temporal fashion, and is defined by patterns of regional and segmental dysfunction in the T2DM heart which are identifiable using machine learning methodologies. Further, machine learning identified the Septal region and AntSeptum segment as locales of interest for therapeutic interventions aimed at ameliorating cardiac dysfunction in T2DM, suggesting that machine learning may provide a more thorough

at https://doi.org/10.5281/zenodo.6391011. We attest that one author had full access to all the data in the study and takes responsibility for its integrity and the data analysis.

**Funding:** This work was supported by the National Institutes of Health from the National Heart, Lung and Blood Institute grant HL128485 and the WVU CTSI grant U54GM104942 awarded to JMH. This work was supported by a National Science Foundation IGERT: Research and Education in Nanotoxicology at West Virginia University Fellowship grant 1144676 awarded to QAH. This work was supported by an American Heart Association Predoctoral Fellowship (AHA 17PRE33660333) awarded to QAH. This work was support by the West Virginia IDeA Network of Biomedical Research WV-INBRE support by National Institute of Health Grant (P20GM103434). This work was supported by the Community Foundation for the Ohio Valley Whipkey Trust awarded to JMH. Imaging experiments and image analysis were performed in the West Virginia University Animal Models & Imaging Facility, which has been supported by the WVU Cancer Institute and NIH grants P20 RR016440, P30 RR032138/GM103488, and S10 RR026378. The funders had no role in study design, data collection and analysis, decision to publish, or preparation of the manuscript.

**Competing interests:** The authors declare that they have no competing interests.

approach to managing contractile data with the intention of identifying experimental and therapeutic targets.

## Introduction

Cardiovascular dysfunction is the leading cause of mortality in the diabetic population, with the risk of developing cardiovascular disease at 2-to-4 times greater than that of the general population [1–3]. As many as 65% of people with diabetes mellitus will die from cardiovascular complications or stroke [1,4]. The identification and diagnosis of cardiac contractile dysfunction, whether sub-clinical or overt, is largely dependent on cardiac imaging modalities such as echocardiography, myocardial perfusion imaging, magnetic resonance imaging, and computed tomography [5]. Conventional echocardiography, including M-mode and pulse wave Doppler (PWD), is the first-line choice for non-invasive diagnosis of cardiac contractile dysfunction, but is limited to the detection of overt systolic dysfunction [5–7].

Speckle tracking echocardiography (STE) can detect sub-clinical changes in cardiac function [8,9], and allows for the observation of cardiac motion as global, regional, or segmental patterns of deformation, making it a useful tool in the diagnosis of cardiac contractile dysfunction. Cardiac remodeling, defined as a change in size, shape, structure, and/or function, is a critical feature of cardiac contractile dysfunction in T2DM and often precedes contractile dysfunction [10–13]. Moreover, left ventricular (LV) wall motion abnormalities, a common consequence of cardiac remodeling, have been linked to 2.4-to-3.4-fold higher risks of cardiovascular dysfunction related mortality [14–16]. STE has the ability to detect these subtle shifts in cardiac function, and is currently utilized clinically in humans and murine models [8,17–20].

Numerous studies have demonstrated the prognostic value of STE, even in patients with no history of cardiac contractile dysfunction. Global longitudinal strain is a widely accepted clinical marker of LV dysfunction, and has been shown to be correlated with diabetic duration in T2DM patients [17,21,22]. Further, STE has been utilized to identify cardiac strain abnormalities and regional afflictions in murine models that may have otherwise been elusive [9,23,24]. Li et al. demonstrated significant reductions in radial and circumferential strains at 16 weeks in *Db/Db* mice, and suggested that strain metrics may be useful for the detection of early LV contractile dysfunction [23,24]. Subsequently, STE may be used to assess localized patterns of dysfunction within the diabetic heart.

Cardiac contractile dysfunction is currently treated via global or LV focused methods regardless of the stage of dysfunction, and the identification of differentially impacted locales in the T2DM heart may provide a modality for pinpointed diagnosis of cardiac contractile dysfunction. At current, spatial (i.e., regional and segmental) alterations have been evaluated in T2DM independently [9,23–26], but the progressive manifestation of regional and segmental cardiac dysfunction in the type 2 diabetic (T2DM) heart remains elusive.

Specifically, current literature addressing cardiovascular dysfunction in diabetes mellitus have at least 1 of the following limitations. Many studies exploring cardiovascular dysfunction were performed in animal models with established T2DM [9,23,27], or human subjects with T2DM [28–33], rather than in populations or individuals at risk of diabetes mellitus and/or cardiovascular dysfunction, with few recent studies focusing on cardiovascular dysfunction in pre-diabetic individuals or individuals at risk of diabetes [34–37]. The limited number of studies that do assess cardiovascular dysfunction prior to the onset of clinically diagnosable

diabetes mellitus find that cardiovascular alterations have already taken place in pre-diabetic individuals, signifying that the development of cardiovascular dysfunction may precede even our earliest timeframe for clinical recognition. Secondly, the primary focus of studies evaluating the use of STE to detect cardiac dysfunction have often focused on STE's ability to detect subclinical cardiovascular dysfunction, or to detect dysfunction earlier than traditional methods [18,25,28–30], rather than its ability to identify and monitor progressive cardiovascular dysfunction over time. Few studies exist which use STE in this manner [24]. Lastly, few studies focus on the nuances of segmental or regional changes in cardiac function, independent of STE and T2DM [24,38–42], with many utilizing aggregate results summarizing the totality of each cardiac suction, such as global longitudinal strain.

Because of the limitations listed above, the progressive manifestation of regional and segmental cardiac dysfunction in the T2DM heart is arguably understudied. Therefore, the objective of this study was to elucidate if machine learning could be utilized to reliably describe patterns of the progressive regional and segmental dysfunction that are associated with the development of cardiac contractile dysfunction in the T2DM heart. We further aimed to utilize machine learning to identity the independent cardiac regions, segments, and features that best describe the cardiac contractile dysfunction in the diabetic heart.

## Materials and methods

All datasets used for machine learning analyses and related code have been made publicly available and can be accessed at https://doi.org/10.5281/zenodo.6391011. We attest that one author had full access to all the data in the study and takes responsibility for its integrity and the data analysis.

### Ethics statement

Animal experiments used in this study conformed to the National Institutes of Health Guidelines for the Care and Use of Laboratory Animals and were approved by the West Virginia University (WVU) Care and Use Committee as an expedited protocol status, and is listed under protocol: 1812377778. Animals were euthanized at 25 weeks of age using cervical dislocation as a primary method, and critical organ removal as a secondary method of euthanasia.

### Experimental animals

Experimental animals included male and female *FVB/NJ* wild-type (WT) mice (RRID: IMSR_JAX:001800) and *FVB/NJ Db/Db* mice (The Jackson Laboratory stock Cat# 006654) [43,44]. *Db/Db* mice develop severe hyperglycemia and obesity between 5 and 6 weeks of age [45]. Mice were housed in the WVU Health Sciences Center animal facility on a 12-hour light/dark cycle in a temperature-controlled room. Animals were maintained on a standard chow diet and had access to both food and water *ad libitum*. No animals were excluded from the study. Initial evaluation of cardiac function in male and female animals presented no significant differences, therefore both sexes were utilized for machine learning analyses.

### Echocardiography

Animals were imaged at 5, 12, 20, and 25 weeks of age. Twelve weeks was chosen to represent a central timepoint between 5 weeks, where initial onset of disease occurs, and 25 weeks, where the diabetic condition is at its most severe. Twenty weeks was chosen as a secondary endpoint due to the potential of animals perishing prior to 25 weeks of age due to the severity of untreated diabetes mellitus and the deterioration caused by the diabetic condition.

Therefore, animals were imaged at both 20 and 25 weeks to ensure that all animals received an echo during the most severe stage of cardiovascular dysfunction. A single trained individual in the WVU Animal Models and Imaging Facility acquired ultrasound images in a blinded fashion in conscious mice to maintain normal left ventricle (LV) function and heart rate [46–49]. Images were acquired using a 32–55 MHz linear array transducer on the Vevo2100 Imaging System (Visual Sonics, Toronto, Canada) as previously described [9,50–52], and were acquired at the highest frame rate (233–401 frames/second) as determined by image resolution. M-mode images were acquired by placing a gate through the center of the short-axis B-mode images to obtain recordings of the internal features of the myocardium. Long-axis B-mode images and short-axis B-mode images were acquired for STE analysis.

## M-mode analysis and pulse-wave doppler

Conventional echocardiography analysis was completed on grayscale M-mode parasternal short-axis images acquired at the mid-papillary level of the LV as previously described [9,50,52]. Data were analyzed by a single trained individual in a blinded fashion. M-mode measurements were calculated over at least three consecutive cardiac cycles and averaged values were considered a single replicate. This was repeated for as many M-mode videos as provided, up to 6 replicates. PWD measurements were acquired by taking at least 3 replicates of consistent cardiac cycles, and calculated in the same manner described above to acquire the best representative measurements. The reliability of conventional measurements was assessed previously [9].

## Speckle tracking strain-based imaging analysis

Velocity, displacement, strain, and strain rate were acquired for all dimensions using the Visual Sonics VevoStrain software (Toronto, Canada). B-mode videos were selected based on the quality of the image and the ability to visualize the endocardial and epicardial wall borders. The borders of the endocardial wall were traced and checked through three consecutive cycles to ensure sufficient tracking. Tracing lines were considered sufficient if they moved faultlessly with both the endocardial and epicardial walls during the three cardiac cycles, ensuring proper measurements. Both the endocardial and epicardial borders were tracked through the image in a frame-by-frame manner. Parameters were calculated in the radial, circumferential, and longitudinal dimensions using the parasternal short and long-axis B-mode videos as previously described [9,23,50,52]. Analysis was performed for both systolic and diastolic values. Velocity, displacement, and strain-based values were initially collected as either positive or negative depending on the direction of motion (i.e., shortening, lengthening, thickening, or thinning). Therefore, values were evaluated as absolute, with both positive and negative values farther from zero indicating faster velocities, increased displacement, increased strain, and increased strain rate. Absolute values were normalized to LV mass. The protocols used to analyze conventional and speckle tracking strain-based echocardiography images, videos, and values can be found at: http://dx.doi.org/10.17504/protocols.io.[14egn2q6qg5d/v1].

## Data cleaning and feature generation

The experimental design and outline used for data collection and the machine learning pipeline are provided in Fig 1. Echocardiography features including conventional echocardiography (i.e., M-mode and PWD), and STE were collected in WT (n = 14) and *Db/Db* (n = 13) mice (Fig 1A). Echocardiography features were documented for 5, 12, 20, and 25 weeks, which included 379 features per timepoint (Fig 1B). Segmental LV features included the following: anterior free segment (AntFree); lateral segment (LatWall); posterior segment (PostWall); inferior free segment (InfFreeWall); posterior septal segment (PostSeptal); and anterior septal

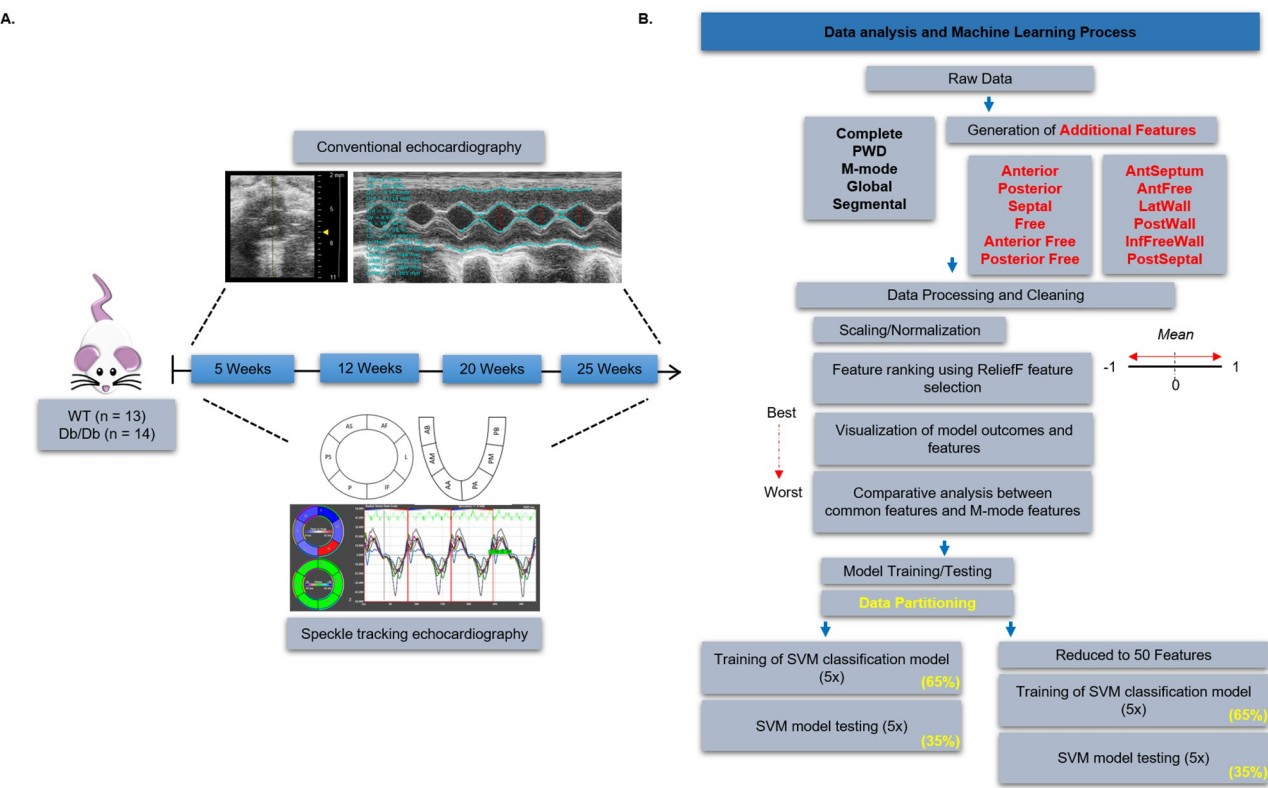

**Fig 1. Schematic of experimental design and the machine learning pipeline.** (A) WT (n = 13) and *Db/Db* mice (n = 14) underwent conventional M-mode and STE assessments at 5, 12, 20, and 25 weeks of age. (B) Data was segregated into 5 raw data sets including both M-mode and STE assessments, as well as 12 datasets generated using STE segmental values. Feature selection and machine learning using a ReliefF algorithm and SVM model were used to determine the reliability of conventional echocardiography as a predictor of T2DM, and further assess the spatial and temporal progression of cardiovascular dysfunction. WT; Wild-type, PWD; pulse-wave doppler, SVM; support vector machine, T2DM; type 2 diabetes mellitus.

segment (AntSeptum). Additional features including the Anterior, Posterior, Septal, Free, Anterior Free, and Posterior Free regions were generated from segmental STE values and included the following: Anterior (AntSeptum, AntFree, and LatWall), Posterior (PostSeptal, InfFreeWall, and PostWall), Septal (AntSeptum and PostSeptal), Free (AntFree, LatWall, Post-Wall, and InfFreeWall), Anterior Free (AntFree and LatWall), and Posterior Free (InfFreeWall and PostWall). The regional features described above were represented by the calculated average of the included segments.

Once all additional features were established, prior to input into MATLAB, the "Complete" datasets, including a total of 379 features per timepoint, were assessed for missing values and outliers. The "Complete" dataset is comprised of all cardiac features, including M-mode, PWD, and STE values. Missing values occurred when specific features were unattainable for a single animal. Outliers were determined as specified in the "*Statistical Analysis*" section of the methods. The number of outliers or missing values within a given feature at a given timepoint typically included two or less values, with rare instances of 3 or more per class (i.e., WT or *Db/Db*). Prior to input for machine learning, outliers and missing values were replaced with the mean value of the feature to negate the loss of animals, maintain model accuracy, and determine features. Missing values and outliers were excluded from statistical analysis using Graph-Pad Prism version 8.02 (GraphPad Prism, RRID:SCR_002798). Each "Complete" dataset was further partitioned into 16 subsets of data containing distinct feature groups: PWD (12

features), M-mode (17 features), Global (36 features), Segmental (195 features), Anterior Free (67 features), Posterior Free (67 features), Anterior (99 features), Posterior (99 features), Septal (67 features), Free (features), AntFree (35 features), LatWall (35 features), PostWall (35 features), InfFreeWall (35 features), PostSeptal (35 features), and AntSeptum (35 features) (Fig 1B). Initial evaluation of normality indicated that data were not consistently normally distributed. Therefore, to account for variability within datasets, non-parametric assessments were performed. Data were normalized using the MATLAB function '*normalize*' to rescale the range of the data to between 0 and 1 to mimic a Gaussian distribution for machine learning applications [53]. Statistical analyses were performed on raw data values without supplemented means for missing data or outliers.

## Feature selection and ranking

In order to estimate feature importance and relevance to binary classification (i.e. WT or *Db/Db*), six feature selection algorithms were assessed for feature ranking capability including: maximum relevancy minimum redundancy, neighborhood component analysis, out-of-bag importance, predictor importance, ReliefF, and chi-square using the MATLAB built-in functions '*fscmrmr*', '*fscnca/featureWeights*', '*fitcencemble/oobPermutedPredictorImportance*', '*fitcencemble/predictorImportance*', '*relieff*', and '*fscchi2*', respectively. The ReliefF MATLAB algorithm, and subsequently the '*relieff*' function, was chosen because it is a method of gaining mutual information about datapoints that is noise-tolerant and able to recognize feature interactions without removing redundant features, thus providing an ideal method to rank features by their importance to binary classification at all timepoints (5, 12, 20, and 25 weeks) [54]. The ReliefF algorithm ranks predictors based on the Euclidean distance, or the length of a distance between two points, for each class. A near-hit was defined as the samples that had a shorter distance from the same class, where a near-miss is represented as a closer sample from another class [54]. The threshold values of near-hit and near-miss were set as the mean values. Finally, the features selected by the ReliefF algorithm were used to assess the effect of feature reduction on testing accuracy in two conditions: 1) above zero score values (i.e., all "relevant" features) and 2) top 50 features most descriptive of cardiac contractile dysfunction. Importantly, the goal of feature selection was not to remove redundant features, but to rank features by their importance for classification in order to manually reduce feature dimensionality for future applications.

## Classification and model performance

Supervised machine learning and data processing was conducted in MATLAB (MathWorks Inc., R2020a, LN: 556683) using the Statistics and Machine Learning Toolbox. To address the relatively small sample size, data was partitioned to training and testing datasets at a 65% training/35% testing ratio using the MATLAB function '*cvpartition*' before 65% of data was used for training. Training data included, at minimum, 9 WT and 8 *Db/Db* mice per testing iteration. A total of 5 testing iterations were performed in which the animals were randomly selected, and the model was trained using five-fold cross validation. The randomization of training data, and cross validation methodologies were used to minimize the risk of model overfitting. Six classification models were initially assessed for training and testing accuracies: classification trees, discriminant analysis, naive bayes, nearest neighbors, support vector machine (SVM), and classification ensembles using MATLAB in-built functions: '*fitctree*', '*fitcdiscr*', '*fitcnb*', '*fitcknn*', '*fitcsvm*', and '*fitcensemble*', respectively. Classification algorithms used echocardiography features to predict the class of the animals based on cardiac function data, placing them into 1 of 2 categories: WT or *Db/Db*. For each dataset, testing was performed for 5 iterations and the average accuracies were obtained for model comparison.

The SVM model, on average, scored the highest accuracy and F-score values between data-sets and was chosen as the highest performing model. Training and testing using the SVM model was performed for 5 iterations before the results were averaged for reporting (S1 Table). Model performance was evaluated using testing accuracies before and after feature reduction. Due to the large size of the "Complete" dataset, we assessed two approaches of reducing dataset dimensionality. The first approach was to reduce feature number to a ReliefF score of greater than zero, where a higher positive score indicated the strength of the feature as an identifier of class, and determines features considered relevant to the classification model. This approach removes irrelevant and redundant features, making it a traditional method of dataset dimensionality reduction. Machine learning was used as the statistical tool to compare the ability of conventional and STE features to identify animals as being within the non-diabetic (WT) or diabetic (*Db/Db*) conditions. Related code is provided on GitHub (https://doi.org/10.5281/zenodo.6391011).

### Most prevalent vs. strongest identifier

Regional and segmental prevalence was determined using the most descriptive of cardiac contractile dysfunction features from each "Segmental" dataset. Each cardiac segment (i.e., AntSeptum, AntFree, LatWall, PostWall, InfFreeWall, and PostSeptal) is represented by a total of 16 features relating to cardiac orientation (i.e., short/systolic, short/diastolic, long/systolic, and long/diastolic) and STE deformation features (i.e., velocity, displacement, strain, and strain rate). Each cardiac region (i.e., Anterior, Posterior, Septal, Free, Anterior Free, Posterior Free) is represented by the number of included segments, multiplied by a factor of 16, to acquire the number of features per segment. "Most prevalent" is used to describe the segment or region with the greatest percentage of features within the top 50 features most descriptive of cardiac contractile dysfunction ranked by their importance for the identification of the diabetic condition. For example, to calculate the prevalence of the Anterior region, each instance of a feature belonging to the segments constructing the Anterior region (AntSeptum, AntFree, and LatWall) were summed and divided by the total number of features selected by the ReliefF algorithm (i.e., 50). Because the number of segments that makes up each region differs, each region was normalized to the number of total contributing segments. The same calculation was applied for segmental prevalence.

"Strongest Identifier" refers to the regions and segments that produced the highest testing accuracies when tested with the machine learning SVM model. The SVM machine learning model was applied to the segmental (i.e., AntSeptum, AntFree, LatWall, PostWall, InfFree-Wall, and PostSeptal) and regional (i.e., Anterior, Posterior, Free, Septal, Anterior Free, Posterior Free) datasets to the determine which segments and/or regions possessed the strongest ability to categorize an animal with cardiac contractile dysfunction as diabetic.

### Statistical analysis

Statistical analyses on raw data values collected during conventional and STE echocardiography were performed using GraphPad Prism version 8.02 (GraphPad Prism, RRID: SCR_002798). Data are presented as mean ± standard error of the mean (SEM). Data visualized in "S1 and S2 Figs" were analyzed using a two-tailed Students T-test. A p-value of $p \leq 0.05$ was considered statistically significant.

## Results

Conventional M-mode echocardiography was utilized to verify progressive cardiac contractile dysfunction in *Db/Db* mice when compared to WT controls. At 5 weeks of age, no significant

**Table 1. Conventional M-mode echocardiography.**

| M-mode | 5 weeks | | 12 weeks | | 20 weeks | | 25 weeks | |
|---|---|---|---|---|---|---|---|---|
| | WT | *Db/Db* | WT | *Db/Db* | WT | *Db/Db* | WT | *Db/Db* |
| Heart rate (BPM) | 698.4 ± 5.8 | 665.3 ± 9.1* | 696.4 ± 11.4 | 667.0 ± 14.3 | 701.8 ± 6.7 | 598.4 ± 27.8* | 684.0 ± 14.7 | 634.7 ± 17.4* |
| EF (%) | 95.8 ± 0.7 | 94.0 ± 0.9 | 92.9 ± 1.0 | 92.6 ± 1.0 | 93.9 ± 0.7 | 88.2 ± 1.5* | 95.0 ± 0.5 | 91.2 ± 0.7* |
| FS (%) | 71.2 ± 1.7 | 67.6 ± 2.0 | 65.2 ± 2.1 | 64.3 ± 1.7 | 66.8 ± 1.4 | 58.5 ± 2.1* | 68.0 ± 1.4 | 60.0 ± 1.7* |
| CO (mL/min) | 10.2 ± 1.2 | 10.1 ± 1.0 | 10.0 ± 0.7 | 11.3 ± 1.1 | 10.3 ± 0.9 | 12.2 ± 0.8 | 9.3 ± 0.8 | 11.1 ± 0.8 |
| SV (µL) | 14.9 ± 1.7 | 15.2 ± 1.5 | 14.3 ± 1.0 | 18.6 ± 1.5* | 14.8 ± 1.3 | 21.1 ± 1.5* | 13.5 ± 1.0 | 19.0 ± 2.0* |
| LV Mass (mm) | 69.4 ± 4.1 | 73.7 ± 3.1 | 79.4 ± 4.2 | 117.7 ± 4.1* | 92.0 ± 5.4 | 134.2 ± 6.9* | 100.1 ± 7.6 | 144.9 ± 6.3* |
| LVAW;s (mm) | 1.8 ± 0.03 | 1.8 ± 0.04 | 1.7 ± 0.07 | 2.0 ± 0.05* | 1.9 ± 0.05 | 2.0 ± 0.04* | 1.9 ± 0.05 | 2.1 ± 0.06* |
| LVAW;d (mm) | 1.1 ± 0.04 | 1.1 ± 0.04 | 1.2 ± 0.04 | 1.3 ± 0.04 | 1.3 ± 0.04 | 1.5 ± 0.04* | 1.3 ± 0.04 | 1.5 ± 0.04* |
| LVPW;s (mm) | 1.8 ± 0.07 | 1.9 ± 0.05 | 1.9 ± 0.05 | 2.2 ± 0.06* | 2.0 ± 0.06 | 2.1 ± 0.09 | 2.1 ± 0.08 | 2.3 ± 0.06 |
| LVPW;d (mm) | 1.3 ± 0.05 | 1.3 ± 0.07 | 1.3 ± 0.09 | 1.7 ± 0.07* | 1.5 ± 0.06 | 1.6 ± 0.09 | 1.6 ± 0.07 | 1.8 ± 0.07 |
| LVED;s (mm) | 0.6 ± 0.04 | 0.7 ± 0.06 | 0.8 ± 0.04 | 0.9 ± 0.06 | 0.7 ± 0.05 | 1.1 ± 0.08* | 0.7 ± 0.04 | 0.9 ± 0.05* |
| LVED;d (mm) | 2.1 ± 0.1 | 2.2 ± 0.09 | 2.1 ± 0.06 | 2.4 ± 0.08* | 2.1 ± 0.08 | 2.6 ± 0.09* | 2.1 ± 0.06 | 2.4 ± 0.1* |
| LVEV;s (mm) | 0.6 ± 0.1 | 1.1 ± 0.3 | 1.2 ± 0.2 | 1.6 ± 0.3 | 1.1 ± 0.2 | 3.3 ± 0.6* | 0.7 ± 0.09 | 1.6 ± 0.2* |
| LVEV;d (mm) | 15.7 ± 1.9 | 16.3 ± 1.7 | 15.6 ± 1.2 | 20.1 ± 1.7* | 15.8 ± 1.5 | 24.4 ± 2.0* | 13.9 ± 0.9 | 19.5 ± 1.9* |

Progressive cardiac dysfunction is observable in Db/Db mice at 12, 20, and 25 weeks of age. Ultrasound images were collected in a blinded fashion in conscious mice to maintain normal LV function and heart rate. "*" denotes significantly different from WT. Data are presented as mean ± SEM. EF; ejection fraction, FS; fractional shortening, CO; cardiac output, SV; stroke volume, LV; left ventricle, LVAW;s; LV anterior wall systolic thickness, LVAW;d; LV anterior wall diastolic thickness, LVPW;s; LV posterior wall systolic thickness, LVPW;d; LV posterior wall diastolic thickness, LVED;s; LV end-systolic diameter, LVED; d; LV end-diastolic diameter, LVEV;s; LV end-systolic volume, LVEV;d; LV end-diastolic volume.

differences in contractile features or LV structural measures were observed in *Db/Db* mice when compared to WT controls (Table 1). At 12 weeks of age, *Db/Db* mice began exhibiting structural changes, including significantly increased LV mass, systolic LV anterior wall thickness (LVAW;s), systolic LV posterior wall thickness (LVPW;s), diastolic LV posterior wall thickness (LVPW;d), and LV end diastolic volume (LVEV;d) and diameter (LVED;d) (Table 1). Additionally, stroke volume (SV) was significantly increased in *Db/Db* mice when compared to WT (Table 1). Together, the 12-week data reveals the development of well-known structural alterations in the diabetic heart. At 20 weeks of age, *Db/Db* mice demonstrated sustained increases in SV, but were further characterized by overt contractile dysfunction marked by significant decreases in ejection fraction (EF) and fractional shortening (FS) when compared to WT controls (Table 1). A similar structural profile was maintained, with the addition of significant increases in diastolic LV anterior wall thickness (LVAW;d) and LV end-systolic volume (LVEV;s), though alterations in posterior wall thicknesses were no longer significant. Cardiac contractile dysfunction and structural changes were maintained at 25 weeks in *Db/Db* mice when compared to WT (Table 1).

To confirm the ability of the ReliefF feature selection algorithm to identify and rank M-mode features by their ability to identify cardiac contractile dysfunction, each of the M-mode data subsets were ranked by their ability to identify cardiac contractile dysfunction at each timepoint (S1 Fig). The 5 echocardiography features identified to be the strongest identifiers of cardiac contractile dysfunction at each timepoint, were tested for statistical significance, and compared against Table 1. Week 5 showed no changes in the 5 best identifiers; heart rate, LVPW;d, LVEV's, EF, or FS (S1A–S1E Fig). At Week 12, the 5 strongest identifiers paralleled significantly altered structural features in Table 1 (S1F–S1J Fig). Four of the 5 features, including LV mass, LVPW;s, LVPW;d, and LVAW;s, were significantly increased in *Db/Db* mice

when compared to WT (S1F–S1I Fig). A single parameter, LVAW;d, was unchanged at 12 weeks (S1J Fig). At week 20, the 5 strongest identifiers included significantly higher LV mass, LVED;s, LVAW;d, and LVEV;s, with significantly lower EF, in *Db/Db* mice when compared to WT (S1K–S1O Fig). At week 25, the 5 strongest identifiers included significantly higher LV mass, LVAW;d, LVEV;s, and LVAW;s, with significantly lower EF, in *Db/Db* mice when compared to WT (S1P–S1T Fig). Together, the 5 strongest identifiers of cardiac contractile dysfunction paralleled the progressive structural changes and decreases in cardiac contractile function observed in Table 1. These data suggest that the ReliefF algorithm was able to adequately rank features based on their importance to the identification of cardiac contractile dysfunction associated with the diabetic condition.

Next, we aimed to determine if conventional echocardiography or STE could more accurately segregate animals into their pre-determined category of diabetic or non-diabetic. An SVM machine learning model was used to compare conventional echocardiography and STE features as identifiers of cardiac contractile dysfunction associated with the diabetic condition. SVM model performance, including training accuracy, standard deviation, testing accuracy, and F-score, were reported for each timepoint in S1 Table. SVM model testing accuracies demonstrated that the "Complete" and Anterior, Septal, and Segmental STE datasets were better able to segregate animals into the correct category of diabetic vs. non-diabetic when compared to conventional echocardiography subsets (M-mode and PWD), Global STE, Free and Posterior STE datasets at all timepoints (Fig 2). At week 5, when no overt systolic dysfunction was detectable, STE datasets including the Septal, Anterior, and Segmental were the strongest identifiers of cardiac dysfunction, while M-mode features were a weak identifier of cardiac dysfunction, and segregated mice with the poorest accuracy (Fig 2). At week 12, the Complete, Segmental, and Anterior datasets were able to identify cardiac dysfunction with the greatest accuracy (Fig 2). By week 20, overt contractile dysfunction was detectable using conventional M-mode echocardiography and PWD, yet the Posterior and Anterior datasets were the strongest identifiers of cardiac dysfunction (Fig 2). At week 25, the STE Free and Global datasets were able to best identify cardiac contractile dysfunction associated with the diabetic condition despite significant decreases in nearly all M-mode features (Fig 2) (Table 1), suggesting that STE echocardiography, which provides strain-based outcomes, may be more capable of distinguishing a state of cardiac contractile dysfunction as compared to changes in systolic function parameters (i.e., EF and FS).

We next aimed to use the ReliefF algorithm to rank features in the "Complete" datasets by their ability to identify a state of cardiac dysfunction. We began with the testing accuracies for the "Complete" dataset, containing 376 features, at all timepoints, and reduced the number of features to only those that were considered to be important for the classification of the diabetic condition. At week 5, the "Complete", unedited, dataset was able to correctly categorize mice as diabetic or non-diabetic 82.0% of the time (Fig 3). Further, the 5 strongest identifiers of cardiac dysfunction were myocardial performance index (MPI), short diastolic (SD) PostSeptal-Wall radial displacement (RD), SD Septal RD, short systolic (SS) AntFree radial strain rate (RSR), and SD InfFreeWall circumferential velocity (CV), which were determined to be significantly different between WT and *Db/Db* mice (Fig 3) (S2A–S2E Fig). Reducing the number of features to those considered "relevant" to the identification of cardiac dysfunction, or having a score above zero, failed to sufficiently reduce dataset size, with the week 5 dataset retaining 158 features (S2 Table). Rather, reducing the complete dataset to only the top 50 strongest identifiers of cardiac contractile dysfunction increased testing accuracy by 14.0% percent as designated by a red arrow (Fig 3) (S3 Table).

At week 12, the "Complete", unedited, dataset was able to correctly categorize mice as diabetic or non-diabetic 96.0% of the time (Fig 3). Further, the 5 strongest identifiers of cardiac

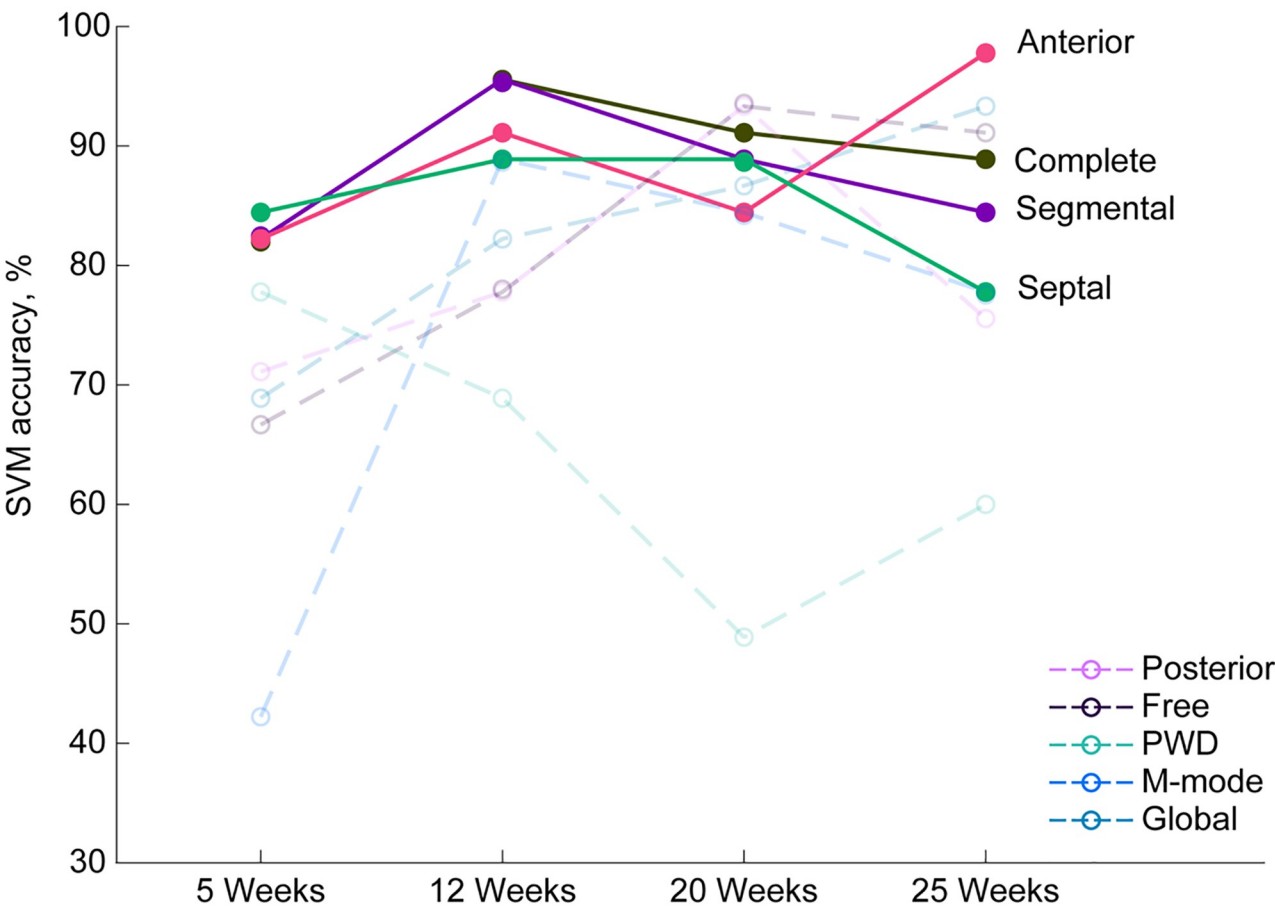

**Fig 2. SVM model testing accuracies.** SVM model testing accuracies were demonstrated for the complete, PWD, M-mode, Global, Segmental, Anterior, Posterior, Septal, and Free datasets at 5, 12, 20, and 25 weeks of age. SVM; support vector machine, PWD; pulse-wave doppler. Values are shown as means ± SEM.

dysfunction were long systolic (LS) AntSeptum radial velocity (RV), LS AntSeptum RSR, SS LatWall CV, LS Septal RSR, and LS Anterior RSR, which were determined to be significantly decreased in *Db/Db* mice when compared to WT (Fig 3) (S2F–S2J Fig). Reducing the number of features to those considered relevant failed to sufficiently reduce dataset size, with the dataset at 12 weeks retaining 319 features (S2 Table). Rather, reducing the complete dataset to only the top 50 strongest identifiers of cardiac contractile dysfunction increased testing accuracy by 2.0% percent as designated by a red arrow (Fig 3) (S3 Table).

At week 20, the "Complete", unedited, dataset was able to correctly categorize mice as diabetic or non-diabetic 91.0% of the time (Fig 3). Further, the 5 strongest identifiers of cardiac contractile dysfunction were LS PostSeptalWall RSR, LS Anterior RSR, LS Septal RSR, SS AntSeptum RSR, and LS Posterior RSR, which were determined to be significantly decreased in *Db/Db* mice when compared to WT (Fig 3) (S2K–S2O Fig). Reducing the number of features to those considered relevant failed to sufficiently reduce dataset size, with the dataset at 20 weeks retaining 330 features (S2 Table). Rather, reducing the "Complete" dataset to only the top 50 strongest identifiers of cardiac contractile dysfunction significantly reduced dataset size, but did not alter testing accuracy (Fig 3) (S3 Table).

At week 25, the "Complete", unedited, dataset was able to correctly categorize mice as diabetic or non-diabetic 89.0% of the time (Fig 3). Further, the 5 strongest identifiers of cardiac

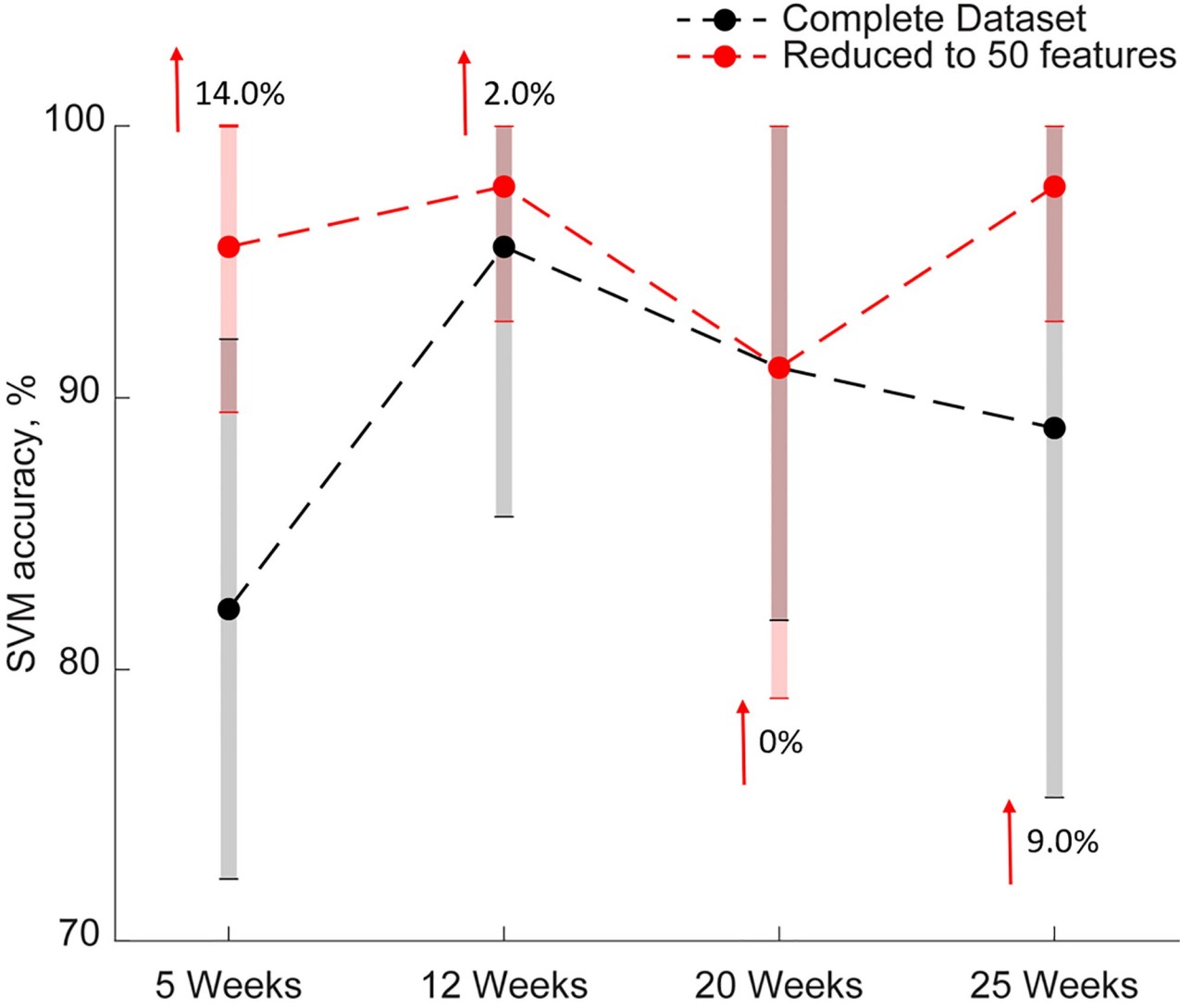

**Fig 3. Machine learning and feature reduction using ReliefF algorithm.** Testing accuracies for the "Complete" dataset at 5, 12, 20, and 25 weeks are shown in black, with a gray bar demonstrating standard deviation of the testing accuracy. Testing accuracies for the "Complete" dataset following feature selection using the ReliefF algorithm are shown in red, with a light red bar demonstrating the standard deviation of the testing accuracy for the top 50 features most descriptive of cardiac contractile dysfunction for the complete dataset. The resulting change in testing accuracy for the reduced dimensionality dataset is shown with a red arrow and the corresponding % increase in accuracy.

dysfunction were SS global radial strain (RS), SS Anterior RS, SS Free RS, SS Anterior RV, and SS AntFree RS, which were determined to be significantly decreased in *Db/Db* mice when compared to WT (Fig 3) (S2P–S2T Fig). Reducing the number of features to those considered relevant failed to sufficiently reduce dataset size, with the dataset at 20 weeks retaining 288 features (S2 Table). Rather, reducing the "Complete" dataset to only the top 50 strongest identifiers of cardiac contractile dysfunction increased testing accuracy by 9.0% as designated by a red arrow (Fig 3) (S3 Table).

Because STE features appeared to identify cardiac dysfunction with the greatest accuracy, the ability of regional and segmental features to identify cardiac contractile dysfunction were assessed using the Segmental, Anterior, Posterior, Septal, Free, Anterior Free, Posterior Free, AntFree, AntSeptum, InfFreeWall, LatWall, PostSeptal, and PostWall datasets. SVM model

performance, including training accuracy, standard deviation, testing accuracy, and F-score, were reported for each dataset at each timepoint in S4 Table. These analyses suggest that changes in regional and segmental features are able to accurately segregate cardiac contractile function associated with the diabetic and non-diabetic conditions.

We next assessed the ability of STE regional and segmental features to identify spatial cardiac dysfunction using two analytical methods. To determine the spatial impact of regional and segmental features on cardiac contractile function, we utilized the Segmental, Anterior, Posterior, Septal, Free, Anterior Free, Posterior Free, AntFree, AntSeptum, InfFreeWall, Lat-Wall, PostSeptal, and PostWall datasets. First, the heart was segregated into regions, which represented locales of the heart that could be evaluated independently from the LV as a whole. Individual cardiac regions were compared against the "Complete" dataset and ranked by their ability to identify cardiac contractile dysfunction, based on how accurately each segregated animals into the diabetic and non-diabetic conditions at each timepoint (Table 2). At week 5, the "Complete" dataset segregated animals into the diabetic and non-diabetic conditions with high accuracy (82.2%), but the Septal region, containing both the AntSeptum and PostSeptal segments, segregated animals as diabetic or non-diabetic with the highest testing accuracy overall (84.4%) (Table 2). At week 12, the "Complete" dataset and the Segmental dataset performed equally well with testing accuracies of 95.6% (Table 2). At week 20, the "Complete" dataset performed with a 91.1% accuracy, while the Free Region, composed of the AntFree, LatWall, PostWall, InfFreeWall, and the Posterior region, composed of the PostSeptal, InfFree-Wall, and PostWall, returned as the strongest independent identifiers of cardiac contractile dysfunction, each with a 93.3% testing accuracy (Table 2). By week 25, the "Complete" dataset performed with an accuracy of 88.9%, much lower than the strongest cardiac region, the Anterior wall, which contained the AntSeptum, AntFree, and LatWall, performed with an overall testing accuracy of 97.8% (Table 2). Because of the variability observed over time, an average accuracy was taken from 5–25 weeks to determine on average, which cardiac regions were best able to identify animals with the diabetic condition. As expected, the "Complete" dataset outperformed individual regions with an average testing accuracy of 89.5%, but three regional datasets stood out as identifying animals with the diabetic condition with high accuracy (≥85%); Anterior (88.9%), Segmental (87.8%), and Septal (85%). These data suggest that these regions are the strongest identifiers of cardiac dysfunction, and may show the greatest level of dysfunction.

The top 4 datasets, performing with average testing accuracies of ≥85% for all timepoints, are highlighted with a bold/italic designation. All values are presented as mean ± standard deviation.

Lastly, cardiac segments which represent the smallest possible locales for evaluation of cardiac function, were ranked by their ability to identify cardiac contractile dysfunction at each timepoint (Table 3). At week 5, the InfFreeWall and the Posterior Free segments were able to best identify animals as having the diabetic condition when compared to other cardiac segments with testing accuracies of 75.6% (Table 3). At week 12, the LatWall and the Anterior Free segments were able to best identify animals as having the diabetic condition with testing accuracies of 97.8% (Table 3). At week 20, the AntSeptum segment was the strongest independent identifier of cardiac contractile dysfunction, and identified animals with the diabetic condition with a 95.6% testing accuracy (Table 3). At week 25, the AntFree and Anterior Free segments were the strongest identifiers of cardiac contractile dysfunction, and identified animals with the diabetic condition with a testing accuracy of 95.6% (Table 3). Overall, three segmental datasets stood out as identifying animals with the diabetic condition with high accuracy (≥85%); LatWall (88.9%), AntSeptum (87.8%), and Anterior Free (87.8%). These data reflect the regions identified above, as all three segments were part of the larger Anterior,

**Table 2. Ranking of regions most representative of overt cardiac contractile dysfunction at 5, 12, 20, and 25 weeks of age.**

| Dataset | %, Test Accuracy | | | | |
|---|---|---|---|---|---|
| | **5 Weeks** | **12 Weeks** | **20 Weeks** | **25 Weeks** | **Average** |
| *Complete* | 82.2 ± 9.94 | 95.6 ± 9.93 | 91.1 ± 9.93 | 88.9 ± 13.61 | 89.5 |
| PWD | 77.8 ± 7.76 | 68.9 ± 21.37 | 48.9 ± 16.85 | 60.0 ± 20.18 | 63.9 |
| M-mode | 42.2 ± 16.48 | 88.9 ± 7.86 | 84.4 ± 12.67 | 77.8 ± 0.00 | 73.3 |
| Global | 68.9 ± 14.49 | 82.2 ± 12.67 | 86.7 ± 18.26 | 93.3 ± 9.94 | 82.8 |
| *Segmental* | 82.2 ± 9.94 | 95.6 ± 6.09 | 88.9 ± 11.11 | 84.4 ± 9.94 | 87.8 |
| *Anterior* | 82.2 ± 9.94 | 91.1 ± 4.97 | 84.4 ± 9.94 | 97.8 ± 4.97 | 88.9 |
| Posterior | 71.1 ± 16.86 | 77.8 ± 0.00 | 93.3 ± 9.94 | 75.6 ± 4.97 | 79.4 |
| *Septal* | 84.4 ± 9.94 | 88.9 ± 7.86 | 88.9 ± 13.61 | 77.8 ± 17.57 | 85.0 |
| Free | 66.7 ± 15.71 | 77.8 ± 7.86 | 93.3 ± 9.94 | 91.1 ± 12.17 | 82.2 |

and Segmental regional datasets, and the AntSeptum being part of the larger Septal regional dataset. Together, these results suggest that these segments are the strongest independent identifiers of cardiac dysfunction, likely driving the regional accuracy observed in Table 2, and may now only show the greatest level of dysfunction, but may be the best independent identifiers cardiac contractile dysfunction associated with the diabetic condition.

The Anterior Free segment contains both the AntFree and LatWall segments, and the Posterior Free segment contains both the InfFreeWall and PostWall segments. The top 3 datasets, performing with average testing accuracies of ≥85% for all timepoints, are highlighted with a bold/italic ***designation***. All values are presented as mean ± standard deviation.

Finally, to determine if the regions and segments defined as being the strongest identifiers of cardiac contractile dysfunction contained the largest number of noteworthy and altered features, we assessed the prevalence of each region and segment. The calculations used to determine the "most prevalent" regions and segments is described in the methods section "*Most Prevalent vs. Strongest Identifier*". At week 5, each region appeared to contribute a similar percentage of features, but the Anterior and Septal regions contributed the largest number of features most descriptive of cardiac contractile dysfunction, representing 19% and 18% of the features, respectively (Fig 4A). Inspection of the prevalence of the individual segments at week 5 revealed that the AntSeptum and InfFreeWall segments were of equal prevalence, with each accounting for 22% of features (Fig 4B). Moreover, the AntSeptum segment was the largest contributor to the prevalence of the Anterior and Septal regions, accounting for the largest percentage of features in both the Anterior and Septal regions at 39% and 61%, respectively (Fig 4C). Accordingly, the InfFreeWall segment also drove the prevalence of the Posterior,

**Table 3. Ranking of segments most representative of overt cardiac contractile dysfunction at 5, 12, 20, and 25 weeks of age.**

| Dataset | %, Test Accuracy | | | | |
|---|---|---|---|---|---|
| | **5 Weeks** | **12 Weeks** | **20 Weeks** | **25 Weeks** | **Average** |
| InfFreeWall | 75.6 ± 12.2 | 82.2 ± 6.1 | 71.1 ± 14.9 | 77.8 ± 11.1 | 76.7 |
| PostWall | 48.9 ± 14.9 | 93.3 ± 9.9 | 88.9 ± 7.9 | 60.0 ± 6.1 | 72.8 |
| *LatWall* | 71.1 ± 12.7 | 97.8 ± 5.0 | 93.3 ± 6.1 | 93.3 ± 9.9 | 88.9 |
| AntFree | 71.1 ± 6.1 | 84.4 ± 18.6 | 71.1 ± 23.0 | 95.6 ± 6.1 | 80.6 |
| *AntSeptum* | 66.7 ± 7.9 | 95.6 ± 9.9 | 95.6 ± 6.1 | 93.3 ± 14.9 | 87.8 |
| PostSeptal | 73.3 ± 16.9 | 75.6 ± 21.4 | 88.9 ± 7.9 | 60.0 ± 12.7 | 74.5 |
| *Anterior Free* | 71.1 ± 16.9 | 97.8 ± 5.0 | 86.7 ± 9.3 | 95.6 ± 6.1 | 87.8 |
| Posterior Free | 75.6 ± 12.2 | 84.4 ± 6.1 | 77.8 ± 11.1 | 91.1 ± 9.3 | 82.5 |

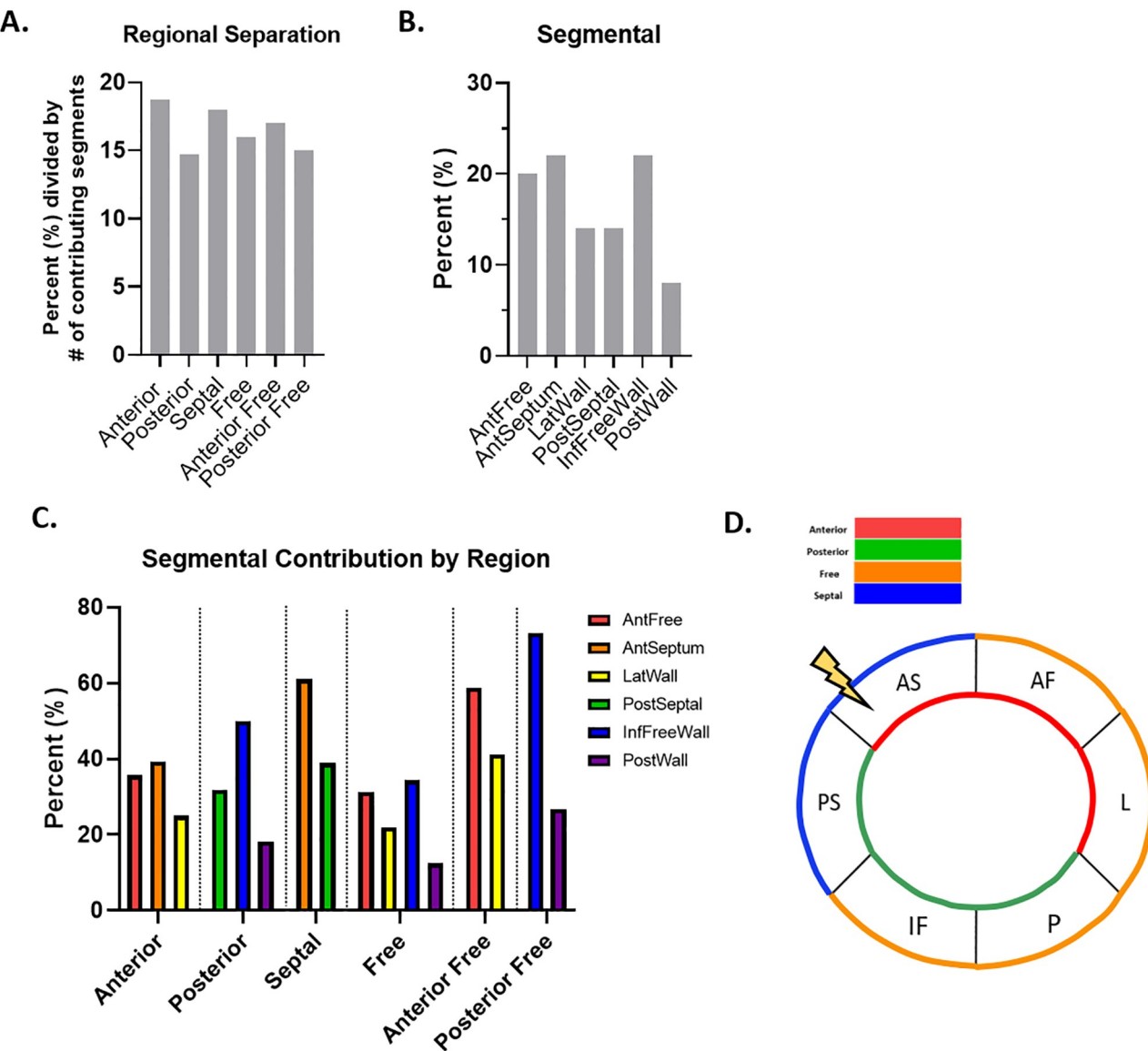

**Fig 4. Spatial and temporal progression of cardiovascular dysfunction at 5 weeks.** (A) Regional separation as calculated with segmental values demonstrates the percent each region is represented in the top 50 features most descriptive of cardiac contractile dysfunction, listed with the regions ranked in order of testing accuracy. (B) The 6 segments were represented as a percentage of all segmental values in the top 50 features most descriptive of cardiac contractile dysfunction, and were ranked in order of testing accuracy. (C) Each region was broken down into its contributing segments relative to the percent contributed by the region. (D) The locale of greatest impact.

Free, and Posterior Free regions (Fig 4C). Based on the number of features contributed by each region and segment, the AntSeptum segment, and the Anterior and Septal regions contained the largest number of noteworthy metrics, which may indicate locales containing the largest number of features contributing to cardiac contractile dysfunction at week 5 (Fig 4D).

At week 12, the Septal region contributed the largest number of features most descriptive of cardiac contractile dysfunction, representing 24% of the features (Fig 5A). Inspection of the prevalence of the individual segments at week 12 further revealed that the AntSeptum segment was of the greatest prevalence, accounting for 28% of the features (Fig 5B). Additionally, the AntSeptum segment was the largest contributor to the prevalence of both the Septal and

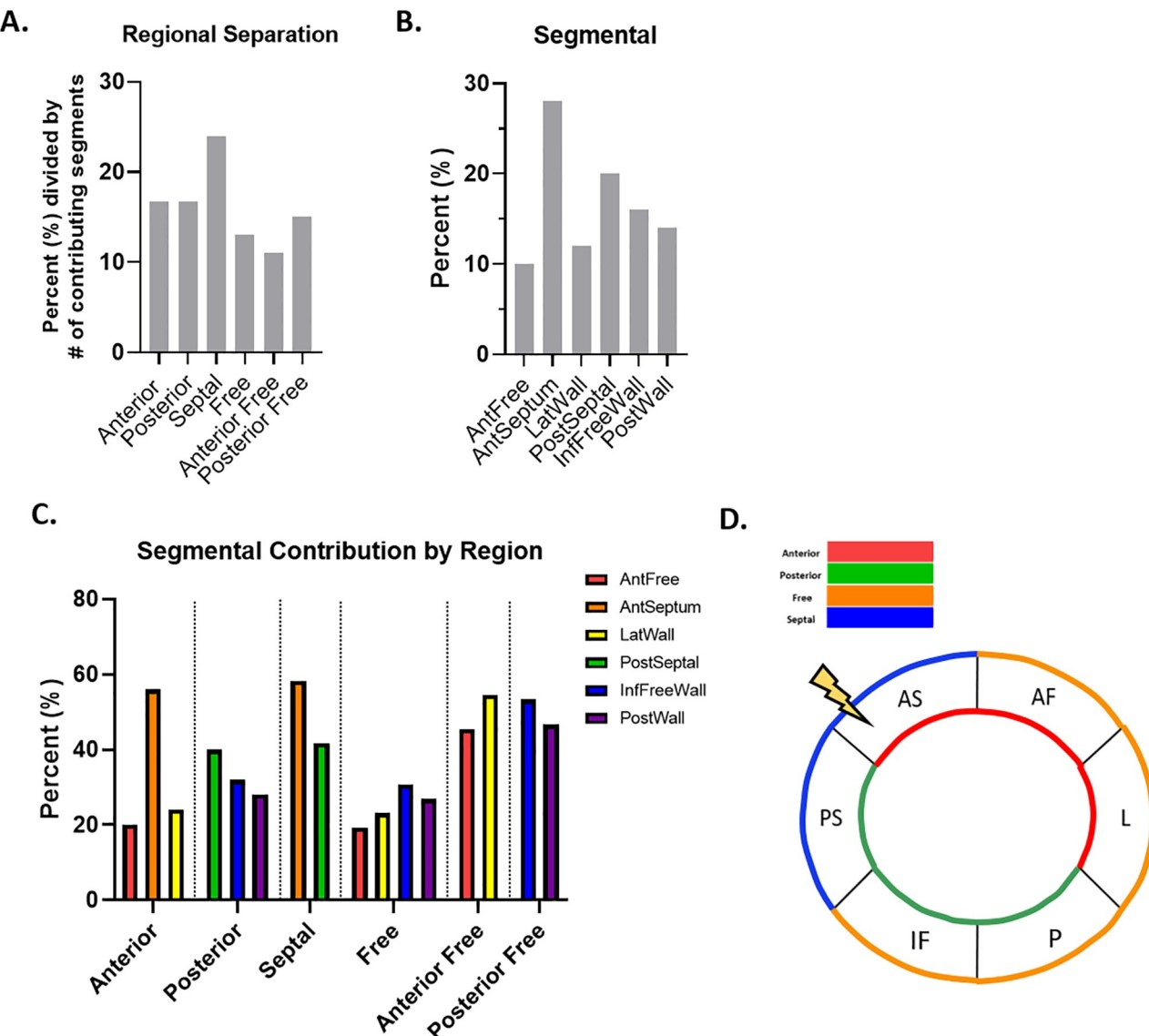

**Fig 5. Spatial and temporal progression of cardiovascular dysfunction at 12 weeks.** (A) Regional representation as calculated with segmental values demonstrates the percent each region is represented in the top 50 features most descriptive of cardiac contractile dysfunction, listed with the regions ranked in order of testing accuracy. (B) The 6 segments were represented as a percentage of all segmental values in the top 50 features most descriptive of cardiac contractile dysfunction, and were ranked in order of testing accuracy. (C) Each region was broken down into its contributing segments relative to the percent contributed by the region. (D) The locale of greatest impact.

Anterior regions, accounting for the largest percentage of features at 58% and 56%, respectively (Fig 5C). Based on the number of features contributed by each region and segment, the AntSeptum segment, and co-concomitantly, the Septal region contained the largest number of noteworthy metrics, and represent the locales containing the largest number of features contributing to cardiac contractile dysfunction at week 12 (Fig 5D).

At week 20, each region contributed a similar percentage of features, but the Anterior and Anterior Free regions contributed the largest number of features most descriptive of cardiac contractile dysfunction, representing 19.3% and 19% of the features, respectively (Fig 6A). Inspection of the prevalence of the individual segments at week 20 demonstrated that the

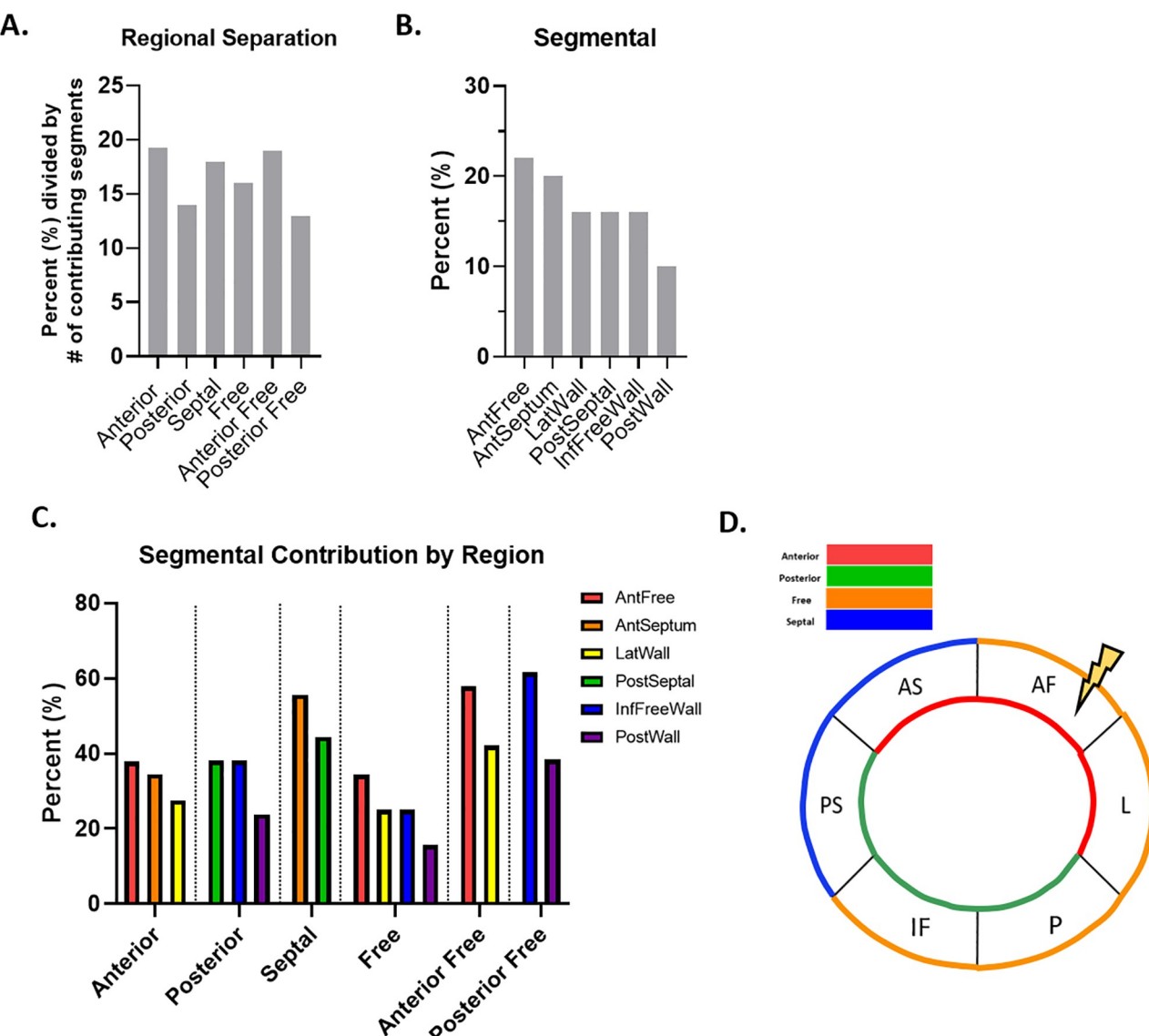

**Fig 6. Spatial and temporal progression of cardiovascular dysfunction at 20 weeks.** (A) Regional representation as calculated with segmental values demonstrates the percent each region was represented in the top 50 features most descriptive of cardiac contractile dysfunction, listed with the regions ranked in order of testing accuracy. (B) The 6 segments were represented as a percentage of all segmental values in the top 50 features most descriptive of cardiac contractile dysfunction and were ranked in order of testing accuracy. (C) Each region was broken down into its contributing segments relative to the percent contributed by the region. (E) The locale of greatest impact.

AntFree segment was the most prevalent, accounting for 22% of features (Fig 6B). The AntFree segment was the largest contributor to the prevalence of the Anterior and Anterior Free regions, accounting for the largest percentage of features in both the Anterior and Anterior Free regions at 38% and 58%, respectively (Fig 6C). Based on the number of features contributed by each region and segment, the AntFree segment, and the Anterior and Anterior Free regions contained the largest number of noteworthy metrics, which may indicate locales containing the largest number of features contributing to cardiac contractile dysfunction at week 20 (Fig 6D).

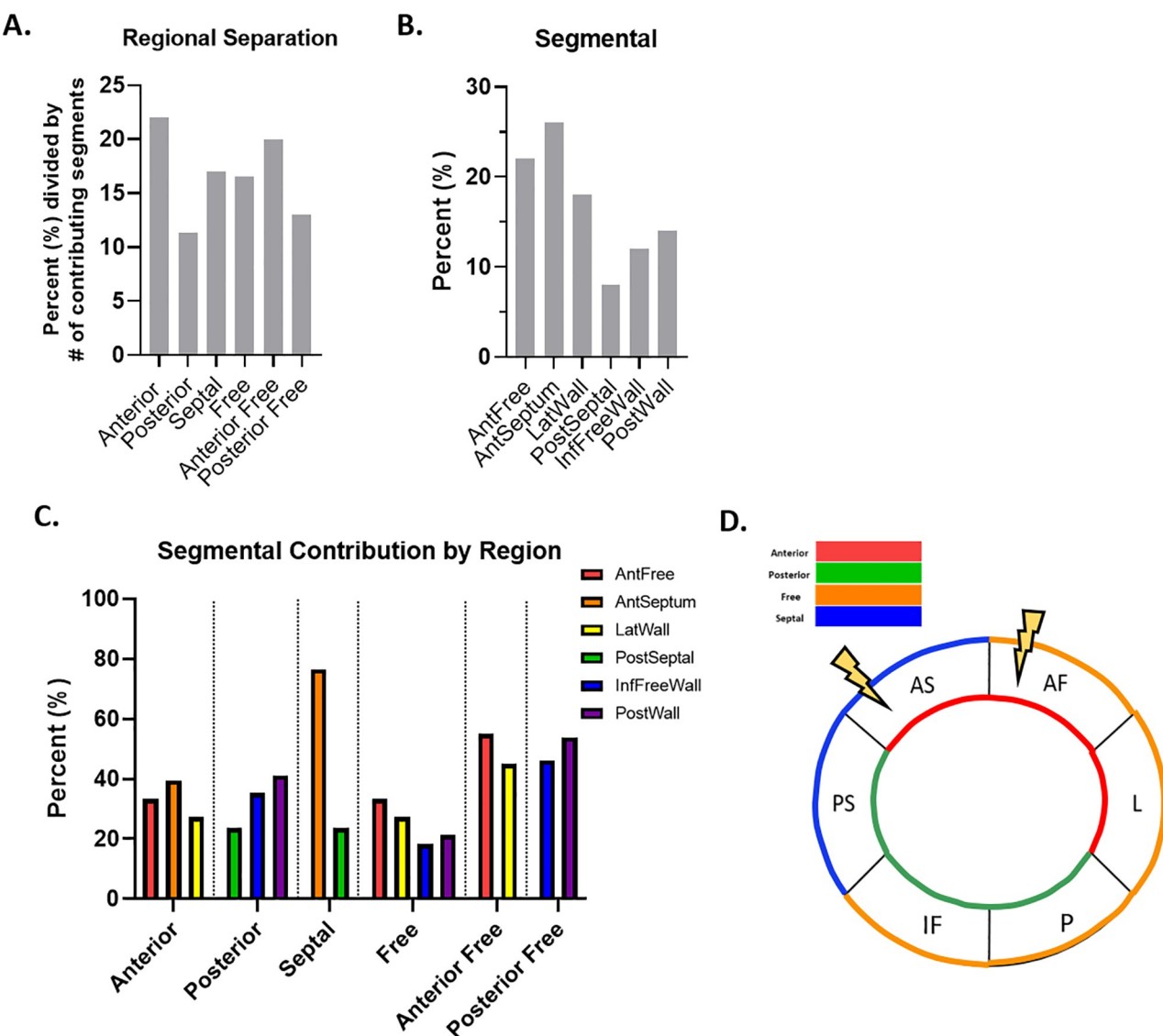

**Fig 7. Spatial and temporal progression of cardiovascular dysfunction at 25 weeks.** (A) Regional representation as calculated with segmental values demonstrates the percent each region was represented in the top 50 features most descriptive of cardiac contractile dysfunction, listed with the regions ranked in order of testing accuracy. (B) The 6 segments were represented as a percentage of all segmental values in the top 50 features most descriptive of cardiac contractile dysfunction and were ranked in order of testing accuracy. (C) Each region was broken down into its contributing segments relative to the percent contributed by the region. (E) The locale of greatest impact.

At week 25, the Anterior and Anterior Free regions contributed the largest number of features most descriptive of cardiac contractile dysfunction, representing 22% and 20% of the features, respectively (Fig 7A). Inspection of the prevalence of the individual segments at week 25 demonstrated that the AntSeptum and AntFree segments were the most prevalent, accounting for 26% and 22% of features, respectively (Fig 7B). The AntSeptum segment was the largest contributor to the prevalence of the Anterior and Septal regions, accounting for the largest percentage of features in both the Anterior and Septal regions at 39% and 76%, respectively (Fig 7C). Based on the number of features contributed by each region and segment, the AntSeptum and AntFree segments, and the Anterior and Anterior Free regions contained the largest

number of noteworthy metrics, which may indicate locales containing the largest number of features contributing to cardiac contractile dysfunction at week 25 (Fig 7D).

## Discussion

The progressive etiology of spatial cardiovascular contractile dysfunction in T2DM remains understudied, and as morbidity and mortality continue to rise, our understanding of its pathophysiology will be critical to produce new and improved diagnostic and treatment opportunities. STE is an invaluable tool for the evaluation of cardiac function, and has been utilized to evaluate changes in contractility and deformation in both murine models of T1DM [9,24], T2DM [23], and human subjects [17,18,22,55]. At current, STE has been utilized in a limited capacity to evaluate progressive cardiac contractile dysfunction in a regional and segmental fashion in the type 2 diabetic heart, and the progressive manifestation of regional and segmental cardiac dysfunction in the T2DM heart remains understudied. Elucidating changes in cardiac function to the fullest extent possible may aid in filling this gap in knowledge, and may provide an alternative method to identify cardiovascular dysfunction in diabetes mellitus patients earlier and with greater specificity than current methods. In this study, we aimed to elucidate if patterns of regional or segmental dysfunction manifest in a progressive fashion in the T2DM heart, in which sub-clinical cardiac contractile dysfunction could be spatially identified and monitored over time. We further aimed to utilize machine learning to identity the cardiac regions, segments, and features that best described a state of cardiac contractile dysfunction using solely non-invasive echocardiography features. A summary of study results and potential applications can be seen in Fig 8.

The use of machine learning enhanced our ability to predict what regions and segments of the heart were most impacted during disease progression, and to further explore those that were best able to identify cardiac contractile dysfunction. Traditional data analyses use descriptive and exploratory methods to provide results and discover patterns in current or past data, but do not make predictions about the future. We aimed to compare traditional data analyses with machine learning methodologies to determine if the regions and segments that were best able to identify progressive cardiac contractile dysfunction also contained the largest number of dysfunctional parameters. By determining the prevalence of a region or segment, we gained insight into the cardiac locales that were likely impacted by T2DM to the greatest extent, and exhibited the largest number of noteworthy changes.

We were able to identify the regions and segments which best identified a state of cardiac contractile dysfunction, and the features which best defined it. The Septal region, and primarily the AntSeptum segment, were determined to be the strongest identifiers of cardiac dysfunction at 5, 20, and 25 weeks. Moreover, the Septal region was identified as a region of interest early in T2DM development and was maintained into the late stages of disease. These data suggest that the Septal region, and the segments contained within, may provide a new metric for the identification of subclinical cardiac dysfunction. The importance of the Septal region may be explained, in part, by the role of the septum in conduction of the heart. The electrical sequence in the heart follows a pre-defined order in healthy individuals, but may be disrupted in individuals with cardiovascular contractile dysfunction [56]. The healthy septum transfers energy between the ventricles, acting as a third pump but diseases that increase septal elastance, such as diabetes mellitus, can resemble left ventricular diastolic dysfunction [57]. Myocardial work, or the contribution by each region to contraction, has been found to be significantly affected by both hypertension and diabetes, with diabetic patients having lower strain values in the septal and lateral segments [58]. Further, the observation of early Septal region dysfunction has been utilized as a method of identifying and monitoring diabetes

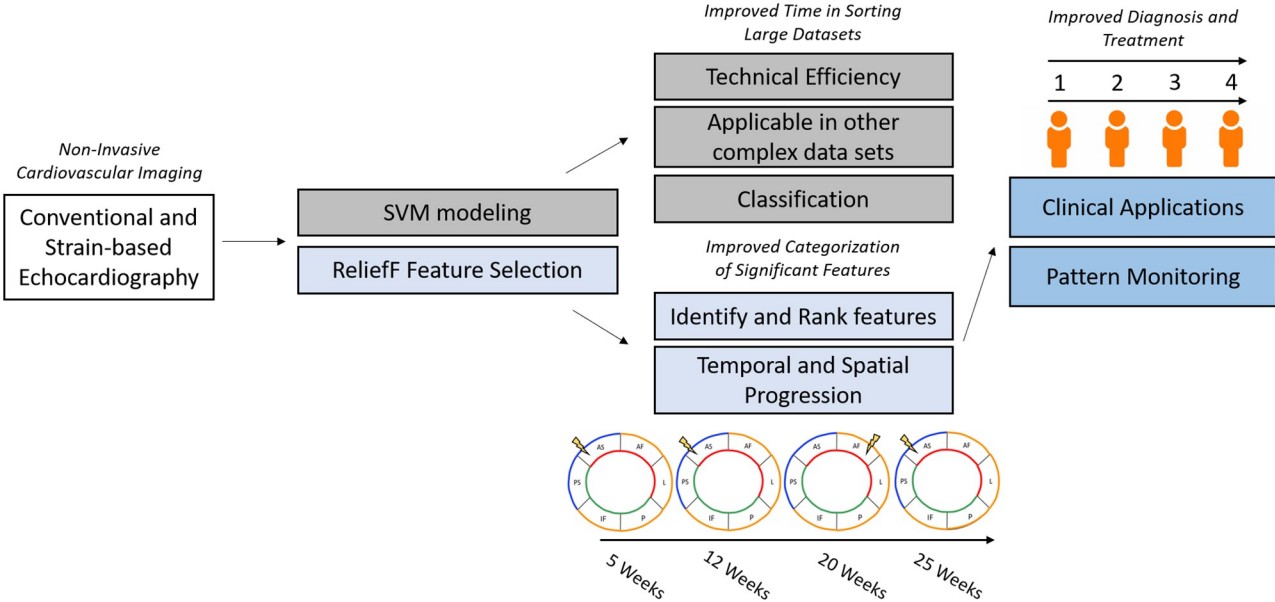

**Fig 8. Summary of methodological contributions and future directions.** Conventional and strain-based echocardiography can be analyzed using machine learning and ReliefF-based feature selection algorithms. Supervised modeling allows for increased technical efficiency and unbiased ranking of features by importance to class prediction, and can be applied in multiple situations concerning large datasets. The ReliefF feature selection algorithm provides an unbiased approach to identify and rank features in order of how important they were for classification and provide insight into the temporal and spatial progression of T2DM, through segmental analysis. The identification of patterns of spatial and temporal features critical to disease progression and development could improve current diagnostic capabilities of cardiovascular contractile dysfunction in T2DM.

mellitus [40,58,59]. These reports suggest that alterations in the Septal region may be observable early in the pathophysiology of T2DM, prior to the onset of clinically recognizable symptoms of cardiac dysfunction. The data presented in this study suggest that the Septal region may benefit most from therapeutic interventions aimed at preventing the progression of cardiac contractile dysfunction in T2DM when initiated early in disease.

Interestingly, the Septal region was the second-best classifier of the diabetic condition at week 12, where the Anterior Free region, and the LatWall segment, were able to identify cardiac contractile dysfunction with the greatest accuracy. This shift from the Septal wall to the AntFree region and LatWall segment at 12 weeks, and back to the Septal region again at 20 weeks, may be the result of the cardiac structural and metabolic remodeling preceding, and ultimately leading to the development of overt contractile dysfunction [60–64]. Metabolic inflexibility and substrate overload initiate several metabolic and structural changes that manifest during the subclinical stages of cardiac contractile dysfunction as an adaptive mechanism to protect the heart [12,61,65,66]. In healthy individuals, the LV primarily performs contractile or shortening work, but in patients with cardiomyopathies, the ability of the septum to provide energy for contraction may be decreased, with a greater amount energy being wasted [67]. This means that even though the Septal region begins the contractile process, the Free and Anterior regions may overcompensate for the Septal regions lack of energy contribution by contributing additional energy for contraction [42]. Metabolically, segments contributing the greatest level of contractile energy also exhibit the highest levels of glucose metabolism, and show a disruption of glucose metabolism, and an ability to produce energy for contraction with disease [38]. This metabolic shift suggests that regions displaying this pattern of energy waste may experience larger amounts of mitochondrial dysfunction, including impaired glucose metabolism, increased reliance on fatty acid oxidation, and changes in mitochondrial

DNA [38,39,68,69]. With this in mind, structural and metabolic alterations may manifest in the Anterior Free region, prior to the onset of overt contractile dysfunction that temporarily improve its ability to identify animals as diabetic or non-diabetic. As a result, we believe that future directions should include the biochemical analysis of regional and segmental metabolism.

Analysis of our data using traditional methods produced results similar to those observed using machine learning. Subsequently, determining the most prevalent regions and segments, or those containing the largest number of noteworthy, and likely dysfunctional metrics were reflective of the most impacted cardiac locale at each timepoint. We observed that the prevalence of regions and segments overlapped, with the most prevalent region containing the most prevalent segment. Overall, these results suggest the AntSeptum segment may contain the greatest number of features contributing to cardiac contractile dysfunction, and may be a metric to identify and monitor throughout the T2DM pathology. Taken together, these data support assessments of regional and segmental function using feature ranking algorithms as a feasible alternative to traditional data analysis to determine regions or locales of interest for experimental and therapeutic purposes.

The healthcare community has largely benefited from the implementation of STE, which has provided a great deal of insight into cardiac contractile dysfunction, and the incorporation of machine learning in the evaluation of echocardiography represents a new and powerful tool for the study [70–72]. Further, machine learning and features ranking methodologies with the intention of identifying regions and segments of interest for experimental and therapeutic purposes. Combining these techniques may provide a more descriptive and thorough approach to managing large amounts of contractile data, as well as improve the process of analyzing and interpreting cardiovascular contractile data [55,73,74]. These applications in echocardiography are increasing exponentially, particularly for their ability to develop innovative models of diagnosis and care [55,75–77]. The initial collection and analysis of data can be difficult, leading to the interpretation of a small subset of data collected, rather than the data as a whole [74]. In clinical settings, the ability to automate data acquisition, processing, and interpretation may help to improve methods of evaluating cardiac dysfunction in the T2DM heart [73,78,79]. The data presented in this study support that machine learning can be used as a tool to identify cardiac contractile dysfunction by using solely non-invasive echocardiography features in a murine model of T2DM. We demonstrate that feature ranking algorithms can be used to identify regional and segmental patterns of cardiovascular contractile dysfunction in T2DM, suggesting that cardiovascular contractile dysfunction occurs not only in a temporal fashion, but progresses spatially.

Additionally, despite significant changes in M-mode parameters, STE outperformed conventional echocardiography at all timepoints, and was consistently better at identifying cardiac contractile dysfunction. Prior to the development of overt systolic dysfunction, as at 5 and 12 weeks, the ability of STE to outperform conventional M-mode echocardiography was expected due to its ability to detect subclinical changes in cardiac function. Alternatively, M-mode echocardiography demonstrated significant decreases in EF and FS features at 20 and 25 weeks of age, but was not as strong of an identifier of cardiac contractile dysfunction as STE features. This discrepancy may be due, in part, to the methodology used by machine learning classification. Specifically, M-mode may contain many significantly altered features, but if the STE regions and segments discussed above contain an overall larger number of altered features, it could indirectly increase the ability of STE features to identify cardiac contractile dysfunction. In terms of clinical applicability, numerous altered STE features may be necessary to outperform the ability of EF to identify contractile dysfunction. Moreover, M-mode echocardiography parameters, including EF and FS, may be a stronger indicator of contractile dysfunction

once overt dysfunction is present, but M-mode echocardiography remains unable to detect clinical and subclinical measures of dysfunction. Hence, the focus of STE should remain the assessment and diagnosis of subclinical cardiac dysfunction. As discussed above, the Septal region may provide a metric for clinicians to identify subclinical changes in cardiac deformation, aid in the diagnosis and staging of cardiac contractile dysfunction prior to the presence of overt systolic dysfunction, and monitor, in addition to EF, during late stages of disease.

It should be noted that animal models are only representative of human disease and may not fully reflect the disease's complex pathology, or traditional onset and progression [80–82]. By design, animal models typically express a pathology of interest in isolation, while many human conditions, including T2DM, occur simultaneously with other disease states [80,82,83]. To achieve this isolation of disease, many of the techniques used to mimic human disease in animal models do not reflect that natural disease progression observed in humans, specifically, genetic alterations or dietary induction of disease in rodent models [80,82–84]. Thus, while animal models are powerful replicators of human disease, further evaluation is necessary to determine if the spatial and temporal patterns of STE observed in the *Db/Db* mouse heart also occur in T2DM human subjects.

## Conclusions

Cardiac contractile dysfunction associated with the T2DM condition manifests spatially, and patterns of regional and segmental dysfunction appear early in the T2DM pathology while progressing in a temporal fashion. Further, the Septal region may provide a metric for the identification of subclinical dysfunction, the diagnosis and staging of cardiac contractile dysfunction prior to the presence of overt systolic dysfunction, and a target for therapeutic interventions aimed at preventing the progression of cardiac contractile dysfunction in T2DM when initiated early in the disease. Additionally, these data support that assessments of regional and segmental function using machine learning and feature ranking algorithms may be a feasible alternative to traditional data analysis and may provide a more descriptive and thorough approach to managing large amounts of contractile data with the intention of identifying regions and segments of interest for experimental and therapeutic purposes.

## Supporting information

**S1 Fig. Top five M-mode features identified by the ReliefF algorithm for each timepoint confirm progression of disease as strong indicators of class.** The 5 echocardiography features identified to be most descriptive of cardiac contractile dysfunction selected for 5 weeks; HR, LVPW;d, LVEV;s, EF, FS (A-E), 12 weeks; LV Mass, LVPW;s, LVPW;d, LVAW;s, LVAW; d (F-J), 20 weeks; LV Mass, LVED;s, EF, LVAW;d, LVEV;s (F-G), and 25 weeks; LV Mass, LVAW;d, EF, LVEV;s, LVAW; s (K-O). HR; heart rate, LV; left ventricle, LVPW;d; LV posterior wall diastolic thickness, LVEV;s; LV end-systolic volume, EF; ejection fraction, FS; fractional shortening, LVPW;s; LV posterior wall systolic thickness, LVED;s; LV end-systolic diameter, LVAW;d; LV anterior wall diastolic thickness. "n" is defined as biological replicates. Figure panels are based in 1 independent experiment. WT and *Db/Db* data were analyzed using a Student's T-test. "*" Denotes $P \leq 0.05$ vs. WT. Values are shown as means ± SEM. (TIF)

**S2 Fig. Analysis in GraphPad of the 5 echocardiography features identified to be most descriptive of cardiac contractile dysfunction for the complete dataset at each timepoint.** (A-E) The 5 echocardiography features identified to be most descriptive of cardiac contractile dysfunction for 5 weeks, (F-J) 12 weeks, (K-O) 20 weeks, and (P-T) 25 weeks. "n" is defined as

biological replicates. Figure panels are based in 1 independent experiment. WT and *Db/Db* data were analyzed using a Student's T-test. "*" Denotes P ≤ 0.05 vs. WT. Values are shown as means ± SEM.
(TIF)

**S1 Table. Performance of supervised machine learning SVM models for all datasets at 5, 12, 20, and 25 weeks of age.** Training accuracies and the associated standard deviations, test accuracies, and F-scores are reported. SVM; support vector machine, PWD; pulse-wave doppler.
(DOCX)

**S2 Table. SVM model performance "relevant" features.** A ReliefF score of above zero was used to select relevant features and reduce dataset dimensionality. Training and testing accuracies are reported for both the full and reduced datasets.
(DOCX)

**S3 Table. SVM model performance of reduced dimensionality dataset containing top 50 features.** The top 50 features were taken and tested as an independent dataset for each time-point. Training and testing accuracies are reported for both the full and reduced datasets.
(DOCX)

**S4 Table. Performance of supervised machine learning SVM models for segmental datasets at 5, 12, 20, and 25 weeks of age.** Training accuracies and the associated standard deviations, test accuracies, and F-scores are reported. SVM; support vector machine.
(DOCX)

## Acknowledgments

We would like to acknowledge the WVU Animal Models and Imaging Facility for their expertise in animal imaging and echocardiography.

## Author Contributions

**Conceptualization:** Andrya J. Durr, Anna S. Korol, Andrew D. Taylor, Saira Rizwan.

**Data curation:** Andrya J. Durr, Quincy A. Hathaway, Andrew D. Taylor, Saira Rizwan, Mark V. Pinti.

**Formal analysis:** Andrya J. Durr, Anna S. Korol.

**Funding acquisition:** John M. Hollander.

**Investigation:** Quincy A. Hathaway, Amina Kunovac.

**Methodology:** Andrya J. Durr, Anna S. Korol, Amina Kunovac.

**Resources:** Quincy A. Hathaway, John M. Hollander.

**Supervision:** Andrya J. Durr.

**Visualization:** Andrya J. Durr, Anna S. Korol.

**Writing – original draft:** Andrya J. Durr.

**Writing – review & editing:** Andrya J. Durr, Anna S. Korol, Amina Kunovac, Andrew D. Taylor, Saira Rizwan, Mark V. Pinti, John M. Hollander.

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
