## [Decision Letter · Decision Letter 0]

20 Mar 2023

PONE-D-23-02008Machine learning for spatial stratification of progressive cardiovascular dysfunction in a murine model of type 2 diabetes mellitusPLOS ONE

Dear Dr. Hollander,

Thank you for submitting your manuscript to PLOS ONE. After careful consideration, we feel that it has merit but does not fully meet PLOS ONE’s publication criteria as it currently stands. Therefore, we invite you to submit a revised version of the manuscript that addresses the points raised during the review process.

We look forward to receiving your revised manuscript.

Kind regards,

Yoshihiro Fukumoto

Academic Editor

PLOS ONE

2. Please make sure that all information entered in the 'Ethics Statement' section regarding ethics approval is also included in the Methods section of the manuscript

3. We noted in your submission details that a portion of your manuscript may have been presented or published elsewhere. [Yes, the week 25 echocardiography assessments in Table 1 are the same week 25 data currently published in the manuscript "Manipulation of the miR-378a/mt-ATP6 regulatory axis rescues ATP synthase in the diabetic heart and offers a novel role for lncRNA Kcnq1ot1". DOI: 10.1152/ajpcell.00446.2021. The inclusion of this data does not constitute dual publication because it is only the same because the same mice were used for both manuscripts, but for two completely separate experiments. We view this overlap as a strategic use of resources, as the related manuscript focuses on echocardiography as a mechanism of testing ejection fraction and fractional shortening, while this submission focuses strictly on the use of echocardiography and machine learning to identify the progressive contractile dysfunction associated with the diabetic condition in a spatial and temporal manner.] Please clarify whether this publication was peer-reviewed and formally published. If this work was previously peer-reviewed and published, in the cover letter please provide the reason that this work does not constitute dual publication and should be included in the current manuscript.

Reviewers' comments:

Reviewer's Responses to Questions

**Comments to the Author**

1. Is the manuscript technically sound, and do the data support the conclusions?

Reviewer #1: Yes

Reviewer #2: Yes

2. Has the statistical analysis been performed appropriately and rigorously? 

Reviewer #1: I Don't Know

Reviewer #2: Yes

3. Have the authors made all data underlying the findings in their manuscript fully available?

Reviewer #1: Yes

Reviewer #2: Yes

4. Is the manuscript presented in an intelligible fashion and written in standard English?

Reviewer #1: Yes

Reviewer #2: Yes

5. Review Comments to the Author

Reviewer #1: Authors examined whether machine learning could be utilized to reliably describe patterns of the progressive regional and segmental dysfunction that are associated with the development of cardiac contractile dysfunction in the T2DM heart. Their study could suggest that that machine learning may provide a more thorough approach to managing contractile data with the intention of identifying experimental and therapeutic targets.

.

Their study seems to be acceptable because their data shows enough data to prove above.

Reviewer #2: This study by Durr et al. was designed to determine whether machine learning could be used to accurately describe patterns of progressive regional and segmental dysfunction associated with the development of cardiac contractile dysfunction in the heart of individuals with T2DM. The authors concluded that cardiac dysfunction in the T2DM heart follows spatial and temporal patterns, which are characterized by regional and segmental dysfunction identifiable through the use of machine learning methodologies.

This study is based on previous publications by the authors and other researchers, although the presented data are preliminary. Therefore, further mechanistic experiments are necessary, especially given recent high-impact publications in PLOS One.

Major comments:

1. In the introduction or discussion, the authors could expand on their mention of speckle tracking echocardiography (STE) and its use in evaluating independent spatial alterations in the diabetic heart. They could provide further details on the limitations of previous studies that have used STE to evaluate cardiac dysfunction in T2DM patients, and explain why the progressive manifestation of regional and segmental cardiac dysfunction in T2DM patients remains understudied.

2. In this study, the researchers used non-invasive echocardiography techniques, including conventional M-mode and Pulse-Wave Doppler, to collect data from mice at specific time points (5, 12, 20, and 25 weeks). They then used advanced machine learning algorithms, specifically support vector machines and ReliefF algorithms, to identify and rank cardiac regions, segments, and features that were most strongly associated with cardiac dysfunction. In simpler terms, the researchers took measurements of the heart's function in mice using ultrasound, and then used computer programs to analyze the data and identify which parts of the heart were most affected by disease. Here, please provide more simple explanations in the Abstract for the authors who are not familiar with the ReliefF algorithms.

3. While the authors did not provide any specific reports in patients with diabetes in the abstract, they suggested that their study provides new insights into the progressive manifestation of regional and segmental cardiac dysfunction in the type 2 diabetic (T2DM) heart. The authors noted that speckle tracking echocardiography (STE) more accurately segregated animals as diabetic or non-diabetic when compared with conventional echocardiography, and that the ReliefF algorithm efficiently ranked STE features by their ability to identify cardiac dysfunction. These findings suggest that STE may be a useful tool in identifying early cardiac dysfunction in patients with T2DM. However, further research is necessary to determine the applicability of these findings in human patients, as animal models may not fully represent the complexities of human disease. Please discuss as to this point.

6. PLOS authors have the option to publish the peer review history of their article (what does this mean?). If published, this will include your full peer review and any attached files.

Reviewer #1: No

Reviewer #2: No

---

## [Author Response · Author response to Decision Letter 0]

20 Apr 2023

Responses to the Editor/Reviewers

Responses to the Editor

If applicable, we recommend that you deposit your laboratory protocols in protocols.io to enhance the reproducibility of your results. Protocols.io assigns your protocol its own identifier (DOI) so that it can be cited independently in the future. For instructions see: https://journals.plos.org/plosone/s/submission-guidelines#loc-laboratory-protocols. 

Additionally, PLOS ONE offers an option for publishing peer-reviewed Lab Protocol articles, which describe protocols hosted on protocols.io. 

Read more information on sharing protocols at https://plos.org/protocols?utm_medium=editorial- email&utm_source=authorletters&utm_campaign=protocols.

Thank you for this opportunity to revise our manuscript, and for this recommendation. We have uploaded the three files necessary to replicate the protocols followed for both conventional and speckle tracking strain-based echocardiography analyses to Protocol.io. as a collection titled “Machine learning for spatial stratification of progressive cardiovascular dysfunction in a murine model of type 2 diabetes mellitus” For reference, the link is provided here: https://www.protocols.io/private/D62464FFD7AA11EDB9470A58A9FEAC02

 and https://journals.plos.org/plosone/s/file?id=ba62/PLOSOne_formatting_sample_title_authors_affiliations.pdf

We have reformatted the manuscript to meet these additional requirements. Please note, the original citations were inserted using EndNote reference manager, and the first author no longer uses this software. All citations, including those for revision, were inserted using Mendeley reference manager and edited to meet the requirements of the Vancouver reference style. 

2. Please make sure that all information entered in the 'Ethics Statement' section regarding ethics approval is also included in the Methods section of the manuscript

We believe that we have corrected this issue. The ethics statements that were listed in the Methods section under “Experimental Animals” have now been moved under a separate section heading within the Methods section titled “Ethics statement” on line numbers 161 – 168 of the “Manuscript” file. 

3. We noted in your submission details that a portion of your manuscript may have been presented or published elsewhere. [Yes, the week 25 echocardiography assessments in Table 1 are the same week 25 data currently published in the manuscript "Manipulation of the miR-378a/mt-ATP6 regulatory axis rescues ATP synthase in the diabetic heart and offers a novel role for lncRNA Kcnq1ot1". DOI: 10.1152/ajpcell.00446.2021. The inclusion of this data does not constitute dual publication because it is only the same because the same mice were used for both manuscripts, but for two completely separate experiments. We view this overlap as a strategic use of resources, as the related manuscript focuses on echocardiography as a mechanism of testing ejection fraction and fractional shortening, while this submission focuses strictly on the use of echocardiography and machine learning to identify the progressive contractile dysfunction associated with the diabetic condition in a spatial and temporal manner.] Please clarify whether this publication was peer-reviewed and formally published. If this work was previously peer-reviewed and published, in the cover letter please provide the reason that this work does not constitute dual publication and should be included in the current manuscript.

Thank you for this clarification. This publication has not been formally reviewed and formally published elsewhere.

1. Has the statistical analysis been performed appropriately and rigorously? Reviewer #1: I Don't Know

Reviewer #2: Yes

We hope that we have further clarified the statistical analyses used and addressed Reviewer 1’s uncertainty. Specifically, we now state that raw data values were collected during conventional and speckle tracking echocardiography on line numbers 380 – 381 of the “Manuscript” file. 

Responses to Reviewers 

Reviewer #1: Authors examined whether machine learning could be utilized to reliably describe patterns of the progressive regional and segmental dysfunction that are associated with the development of cardiac contractile dysfunction in the T2DM heart. Their study could suggest that that machine learning may provide a more thorough approach to managing contractile data with the intention of identifying experimental and therapeutic targets.

.

Their study seems to be acceptable because their data shows enough data to prove above.

Reviewer #2: This study by Durr et al. was designed to determine whether machine learning could be used to accurately describe patterns of progressive regional and segmental dysfunction associated with the development of cardiac contractile dysfunction in the heart of individuals with T2DM. The authors concluded that cardiac dysfunction in the T2DM heart follows spatial and temporal patterns, which are characterized by regional and segmental dysfunction identifiable through the use of machine learning methodologies.

This study is based on previous publications by the authors and other researchers, although the presented data are preliminary. Therefore, further mechanistic experiments are necessary, especially given recent high-impact publications in PLOS One.

Major comments:

1. In the introduction or discussion, the authors could expand on their mention of speckle tracking echocardiography (STE) and its use in evaluating independent spatial alterations in the diabetic heart. They could provide further details on the limitations of previous studies that have used STE to evaluate cardiac dysfunction in T2DM patients, and explain why the progressive manifestation of regional and segmental cardiac dysfunction in T2DM patients remains understudied.

Thank you for your comment. In the Introduction, we have expanded on the limitations of previous studies assessing speckle tracking echocardiography in T2DM patients. Briefly, we discuss three limitations: many studies exploring cardiovascular dysfunction were performed in animal models with established T2DM or human subjects with T2DM, rather than in populations or individuals at risk of diabetes mellitus and/or cardiovascular dysfunction, the primary focus of studies evaluating the use of STE to detect cardiac dysfunction have often focused on STE’s ability to detect subclinical cardiovascular dysfunction rather than its ability to identify and monitor progressive cardiovascular dysfunction over time, and few studies focus on the nuances of segmental or regional changes in cardiac function, with many utilizing aggregate results summarizing the totality of each cardiac suction, such as global longitudinal strain. These additions can be found on line numbers 126 – 143 of the “Manuscript” file. Through this discussion, we hope we have sufficiently explained why the progressive manifestation of regional and segmental cardiac dysfunction in T2DM patients is understudied. 

2. In this study, the researchers used non-invasive echocardiography techniques, including conventional M-mode and Pulse-Wave Doppler, to collect data from mice at specific time points (5, 12, 20, and 25 weeks). They then used advanced machine learning algorithms, specifically support vector machines and ReliefF algorithms, to identify and rank cardiac regions, segments, and features that were most strongly associated with cardiac dysfunction. In simpler terms, the researchers took measurements of the heart's function in mice using ultrasound, and then used computer programs to analyze the data and identify which parts of the heart were most affected by disease. Here, please provide more simple explanations in the Abstract for the authors who are not familiar with the ReliefF algorithms.

Thank you for your comment. We altered the abstract to include a description of both the support vector machine model, which classifies data using a single line, or hyperplane, that best separates each class, and the ReliefF algorithm, which ranks features by how well each feature lends to the classification of data. (line numbers 66 – 68 of the “Manuscript” file). We believe this addition will ensure that readers have the necessary context on the primary machine learning algorithms used to understand the study.

3. While the authors did not provide any specific reports in patients with diabetes in the abstract, they suggested that their study provides new insights into the progressive manifestation of regional and segmental cardiac dysfunction in the type 2 diabetic (T2DM) heart. The authors noted that speckle tracking echocardiography (STE) more accurately segregated animals as diabetic or non-diabetic when compared with conventional echocardiography, and that the ReliefF algorithm efficiently ranked STE features by their ability to identify cardiac dysfunction. These findings suggest that STE may be a useful tool in identifying early cardiac dysfunction in patients with T2DM. However, further research is necessary to determine the applicability of these findings in human patients, as animal models may not fully represent the complexities of human disease. Please discuss as to this point.

Thank you for your comment. We agree that this is a very important point of discussion and have added text to the discussion describing this topic. Specifically, we address the notion that animal models may not fully represent the complexities of human disease as mentioned in your comment, and discuss several reasons why what we observe in animal models may or may not be reflected in human subjects. This addition can be found on line numbers 868 – 877 in the “Manuscript” file.

Additional response

Figs 1-8 were uploaded to PACE and a PACE corrected version was downloaded, reviewed, and submitted with the revised manuscript.

---

## [Decision Letter · Decision Letter 1]

25 Apr 2023

Machine learning for spatial stratification of progressive cardiovascular dysfunction in a murine model of type 2 diabetes mellitus

PONE-D-23-02008R1

Dear Dr. Hollander,

We’re pleased to inform you that your manuscript has been judged scientifically suitable for publication and will be formally accepted for publication once it meets all outstanding technical requirements.

Kind regards,

Yoshihiro Fukumoto

Academic Editor

PLOS ONE

Additional Editor Comments (optional):

Reviewers' comments:

Reviewer's Responses to Questions

**Comments to the Author**

1. If the authors have adequately addressed your comments raised in a previous round of review and you feel that this manuscript is now acceptable for publication, you may indicate that here to bypass the “Comments to the Author” section, enter your conflict of interest statement in the “Confidential to Editor” section, and submit your "Accept" recommendation.

Reviewer #2: All comments have been addressed

2. Is the manuscript technically sound, and do the data support the conclusions?

Reviewer #2: Yes

3. Has the statistical analysis been performed appropriately and rigorously? 

Reviewer #2: Yes

4. Have the authors made all data underlying the findings in their manuscript fully available?

Reviewer #2: Yes

5. Is the manuscript presented in an intelligible fashion and written in standard English?

Reviewer #2: Yes

6. Review Comments to the Author

Reviewer #2: The authors improved the manuscript according to the Reviewer's comments. There is no further comment for this Reviewer.

7. PLOS authors have the option to publish the peer review history of their article (what does this mean?). If published, this will include your full peer review and any attached files.

Reviewer #2: No

---

## [Editor Report · Acceptance letter]

28 Apr 2023

PONE-D-23-02008R1 

Machine learning for spatial stratification of progressive cardiovascular dysfunction in a murine model of type 2 diabetes mellitus 

Dear Dr. Hollander:

I'm pleased to inform you that your manuscript has been deemed suitable for publication in PLOS ONE. Congratulations! Your manuscript is now with our production department. 

Kind regards, 

on behalf of

Dr. Yoshihiro Fukumoto 

Academic Editor

PLOS ONE